# Semiconductor-less vertical transistor with $I_{ON}/I_{OFF}$ of $10^6$

Jun-Ho Lee[1,8], Dong Hoon Shin [2,8], Heejun Yang[3,8], Nae Bong Jeong[1], Do-Hyun Park[1], Kenji Watanabe [4], Takashi Taniguchi [5], Eunah Kim[6], Sang Wook Lee[2], Sung Ho Jhang[1], Bae Ho Park [1], Young Kuk[7] & Hyun-Jong Chung [1✉]

Semiconductors have long been perceived as a prerequisite for solid-state transistors. Although switching principles for nanometer-scale devices have emerged based on the deployment of two-dimensional (2D) van der Waals heterostructures, tunneling and ballistic currents through short channels are difficult to control, and semiconducting channel materials remain indispensable for practical switching. In this study, we report a semiconductor-less solid-state electronic device that exhibits an industry-applicable switching of the ballistic current. This device modulates the field emission barrier height across the graphene-hexagonal boron nitride interface with $I_{ON}/I_{OFF}$ of $10^6$ obtained from the transfer curves and adjustable intrinsic gain up to 4, and exhibits unprecedented current stability in temperature range of 15–400 K. The vertical device operation can be optimized with the capacitive coupling in the device geometry. The semiconductor-less switching resolves the long-standing issue of temperature-dependent device performance, thereby extending the potential of 2D van der Waals devices to applications in extreme environments.

[1] Department of Physics, Konkuk University, Seoul, Republic of Korea. [2] Department of Physics, Ewha Womans University, Seoul, Republic of Korea. [3] Department of Physics, Korea Advanced Institute of Science and Technology, Daejeon 34141, Republic of Korea. [4] Research Center for Functional Materials, National Institute for Materials Science, Tuskuba, Japan. [5] International Center for Materials Nanoarchitectonics, National Institute for Materials Science, Tuskuba, Japan. [6] Department of Energy Science, Sungkyunkwan University, Suwon, Republic of Korea. [7] Daegu Gyeongbuk Institute of Science & Technology, Daegu, Republic of Korea. [8] These authors contributed equally: Jun-Ho Lee, Dong Hoon Shin, Heejun Yang. ✉email: hjchung@konkuk.ac.kr

Semiconductors have been indispensable to solid-state electronic devices since the first solid-state electronic device (i.e., the transistor in 1947) because the channel current of the transistor must be modulated by the carrier (electron and hole) density, which relies on the bandgap of the semiconductors[1]. With the rapid development of the semiconductor industry, conventional three-dimensional (3D) semiconductors (Si, GaAs, and InP) are encountering challenges in terms of increasing further spatial resolution of the device and temperature-dependent device performances in various environments. The enhanced electric field degrades the carrier mobility in the semiconductor channel. It is because, with the enhanced electric field, the carrier starts to scatter with optical phonon of the semiconductors and lose more of its energy, resulting in the velocity saturation[1]. Also, the carrier density or device performance depends on the temperature and, as a result, deviates from Moore's law[2,3]. To overcome the first challenge, vacuum-channeled devices (vacuum field-effect transistors), which resemble the primitive vacuum tube triode of the early 1900s, have attracted enormous interest because they utilize ballistic transport by tunneling through the vacuum channel[4], and recently, they demonstrate the long-term stability processability in 150-mm water scale[5]. However, industry-applicable current switching was not realized in the tunneling devices, and the source (e.g., silicon) and gate (e.g., indium tin oxide) currents continue to rely on thermally generated carriers, which retain most of the drawbacks associated with conventional semiconductor devices.

As an alternative, two-dimensional (2D) vertical device structures have been proposed[6–9]. Despite its unprecedentedly high room temperature mobility[10], the graphene FET (GFET) still suffers from insufficient switching ($I_{ON}/I_{OFF}$ ~ 10 at room temperature) because of the absence of a bandgap[11]. Additionally, we know that artificial bandgap opening in graphene inevitably sacrifices the mobility[12]. In contrast, transition metal dichalcogenide-based FETs have shown $I_{ON}/I_{OFF}$ values of up to $10^8$ using their bandgaps, but their carrier mobilities remain at ~20% of that of Si[7]. These inherent limitations can be resolved by using vertical van der Waals heterostructures and work function modulation of graphene as a switching principle[13]. This principle was originally demonstrated in graphene barristors (GBs)[14] and has been used in various devices containing either organic[15–18] or inorganic[19–25] semiconductor–graphene junctions. In addition, bipolar junction transistor-like devices have been also investigated, where graphene was used as a base material, thus called as graphene-base transistor[26–30]. The switching in such devices does not rely on the thermally generated charge of semiconductors, but semiconductors are still crucial elements required to achieve efficient switching. Thus these 2D devices have the same limitations as conventional semiconductor devices: scattering-limited carrier mobility and temperature-dependent device performance.

The ideal solution (i.e., the effective switching of ballistic transport without semiconductors) has not yet been realized; indeed, only one-order modulation of $I_{ON}/I_{OFF}$ has been reported[31]. To control ballistic transport adequately, we considered two modes by which current can tunnel through either vacuum or insulator channels: (1) direct tunneling (DT), which most graphene tunneling devices (including ref. [31]) use for switching, and (2) field emission (FE), which has been rarely explored. The DT is proportional to the density of states (DOS) of two electrodes, whereas the FE is exponentially influenced by the tunneling barrier height[32]. When the electric field modulates the charges on graphene, both the work function and DOS at the Fermi level of the graphene are modulated. However, the two tunneling-current modes behave differently under modulation. Although the DT current produces physically limited insufficient switching (e.g., an $I_{ON}/I_{OFF}$ of ~50 at room temperature) via the DOS modulation of

graphene[33], the FE current can be largely modulated by an exponential function of the barrier height.

Here we report a semiconductor-less electronic device based on a van der Waals vertical heterostructure of metal–hexagonal boron nitride ($h$BN)–graphene–$h$BN–metal (Fig. 1a–b). We selected the stacked structure as the platform for an FE tunneling current because the graphene–$h$BN junction is the cleanest 2D semimetal–insulator system[34]. While a vacuum could be another candidate for the tunneling barrier of semiconductor-less devices without a dielectric breakdown, it would require a higher operating voltage to overcome the vacuum's barrier height to switch the current. The device mainly switches the FE current by modulating the FE barrier height; therefore, we termed the device a "field-emission barristor" (FEB; Fig. 1c–f). Based on the exponential barrier height dependence of the FE current, we achieved an $I_{ON}/I_{OFF}$ of up to $10^6$ without using semiconductors (Fig. 1g). Consequently, the switching performance of our FEB exhibited ignorable degradation at 15 K (Fig. 1h–j), a temperature at which conventional semiconductor devices cannot operate. We calculated the FE barrier height variation by work function modulation in graphene using Fowler–Nordheim (FN) plot. Moreover, the work function modulation in graphene is reliably manipulated by the capacitive coupling among the gate capacitance ($C_{Gate}$), tunneling-channel capacitance ($C_{TC}$), and quantum capacitance of graphene ($C_Q$) in the FEB. Consequently, the above coupling effect is universal in all 2D vertical device geometries, which implies that the optimization principle can be applied to other vertical devices to improve their performance.

## Results

**Transport characteristics of semiconductor-less transistor.** In Fig. 1g, the FEB with a gate-$h$BN thickness ($t_{Gate}$) of 62 nm and a tunneling-$h$BN thickness ($t_{TC}$) of 64 nm shows an efficient switching (by an $I_{ON}/I_{OFF}$ of up to $10^6$) without semiconductors. The channel current ($I_D$) increases exponentially by an increase of the gate bias ($V_G$). As the $V_G$ increases, more electrons are accumulated on the graphene, which decreases the work function of graphene by the square root of the electron density and the tunneling barrier height ($\Phi_B$) for the "on" state. The FE current $I_D$, which increases exponentially as the $\Phi_B$ decreases, can be described as follows[35]:

$$I(V) = \frac{A_{eff}}{8\pi h \Phi_B d^2 m^*} \frac{q^3 m V^2}{} \exp\left[\frac{-8\pi\sqrt{2m^*}\Phi_B^{\frac{3}{2}}d}{3hqV}\right], \qquad (1)$$

where $A_{eff}$ is the effective tunneling area, $q$ is the elementary charge, $m$ is the mass of electron or hole, $m^*$ is the tunneling effective mass, $V$ is the applied voltage, $h$ is Plank's constant, and $d$ is the tunneling distance. While the Schottky current depends on the temperature, the FE current barely depends on the temperature[36]. The above formula supports the critical device operation, the switching with an $I_{ON}/I_{OFF}$ ratio of ~$10^6$, presented in Fig. 1g. The performance is unique among graphene-based logic devices without semiconductors. Indeed, former graphene-based tunneling or lateral devices based on DOS-dependent channel current have $I_{ON}/I_{OFF}$ ratios of ~10, which is a physical limit imposed by the fact that the charge density modulation is limited to 100 at room temperature[37]. As $V_G$ increases, the tunneling mechanism of the electrons for the gate leakage current is changed from DT to FE at $V_G = 14$ V or a gate field of 0.23 V/nm in a similar manner with $I_D$. The $I_G$ remains <0.5% of $I_D$ in the $V_G$ range. Minimizing the leakage effect on the $I_D$, the gate field was limited to 0.23 V/nm in our study.

A critical issue affecting semiconductor-based devices—i.e., temperature-limited operation—can be resolved by our

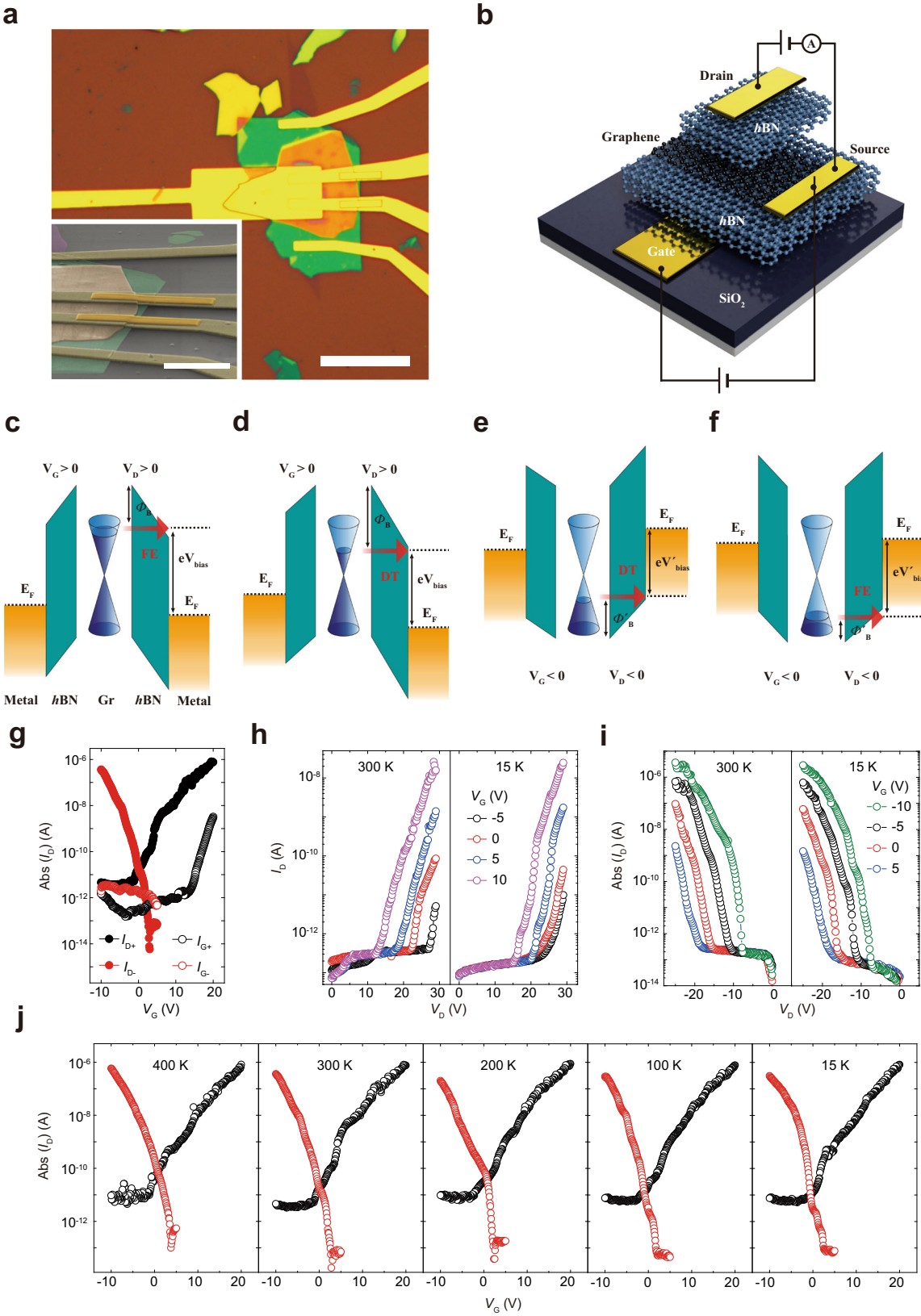

semiconductor-less ballistic device. Figure 1h–j show the temperature-independent performance of the FEB: the channel current ($I_D$) exhibits little variation at temperatures of 15–400 K under various operating conditions. This independence is attributable to the nature of the FE tunneling. Notably, the current does not degrade even at $T = 15$ K, at which the charge carriers of most semiconductors would be frozen[2]. The absence of degradation is a characteristic feature of our semiconductor-less ballistic device.

The channel current in Fig. 1h, i shows two domains that reflect two different tunneling mechanisms (DT and FE) depending on the drain voltage ($V_D$). First, ineffective gating

**Fig. 1 Fabrication of the FEB and its semiconductor-less device characteristics. a** Optical microscopic image of an FEB consisting of stacked metal/$h$BN/graphene/$h$BN/metal (scale bar 20 μm). (inset) Scanning electron microscopic image of polymethyl methacrylate (PMMA) bridges, which help thin metal electrodes connect through the thick stack (scale bar: 5 μm) (for more detail, see the "Methods" section). **b** Schematic diagram of the FEB applying $V_D$ and $V_G$. **c**, **d** Band diagrams of the FEB ($V_D > 0$) under FE-dominant (**c**) and DT-dominant conditions (**d**). The $V_G$ modulates the accumulation of electrons on the graphene. **e**, **f** Band diagrams of the FEB ($V_D < 0$) under DT-dominant (**e**) and FE-dominant conditions (**f**). The $V_G$ modulates the accumulation of holes on the graphene. The gate voltage decreases from **c** to **f**. **g** $I_D$ switching performance of the semiconductor-less device. $I_{D-}$ and $I_{G-}$ are drain and gate current under $V_D = -18$ V. $I_{D+}$ and $I_{G+}$ are drain and gate current under $V_D = 29$ V. For the negative $V_G$, $I_{ON}/I_{OFF}$ above $10^6$ has been achieved at 300 K. **h** Device characteristics of the n-type FEB ($V_D > 0$) at 300 K (left) and 15 K (right). **i** Device characteristics of the p-type FEB ($V_D < 0$) at 300 K (left) and 15 K (right). For both types, very little temperature degradation of $I_D$ was observed. **j** $I_D$ switching under $V_D = -18$ V (red) and 29 V (black) also exhibited little temperature degradation from 400 to 15 K.

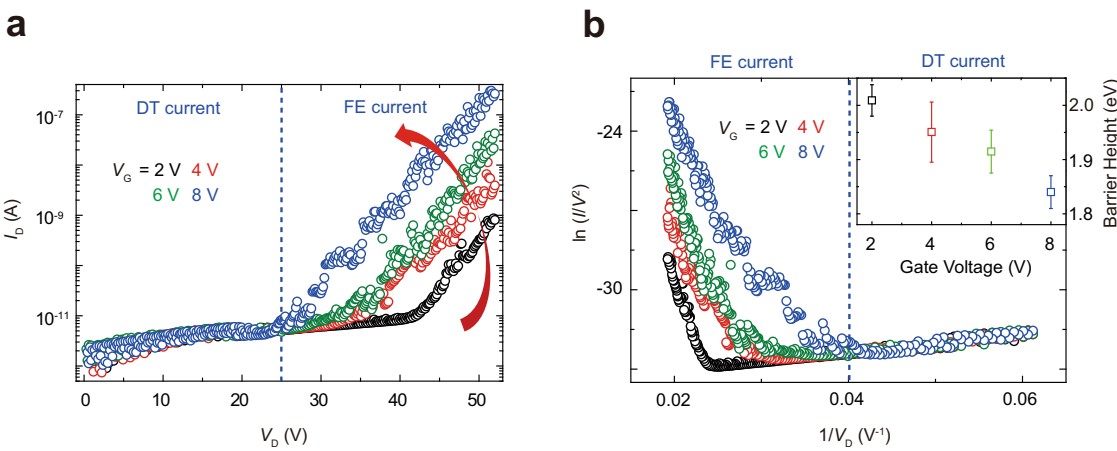

**Fig. 2 Single-emitter approximation of the FE from graphene. a** Output characteristics of FEB were measured by varying $V_D$ from 0 to 52 V and $V_G$ from 2 to 8 V. As $V_G$ increases, turn-on voltage decreases because graphene's Fermi-level increases (barrier height decreases). **b** The characteristics were replotted with axes of $\ln(I/V^2)$ and $1/V$. From the linear fitting of the lower part (blue dashed line for $V_G = 2$ V), barrier heights were extracted to 2.01, 1.95, 1.91 and 1.84 eV when $V_G = 2$, 4, 6 and 8 V, respectively (error bars represent standard error). The height decreases by 0.17 eV, as $V_G$ increases from 2 to 8 V.

($I_{ON}/I_{OFF} \sim 10$) appears in the DT regime at low $V_D$, whereas effective gating ($I_{ON}/I_{OFF} > 10^4$) is activated at high $V_D$. The increase in the drain voltage converts the channel current from DT to FE, allowing the modulation of the FE current shown in Fig. 1j. The transition voltages from DT to FE under positive $V_D$ and negative $V_D$ decrease from 27 V ($V_G = -5$ V) to 13 V ($V_G = 10$ V) and from $-19$ V ($V_G = 5$ V) to $-8$ V ($V_G = -10$ V), respectively; thus a higher $V_G$ realizes a lower $\Phi_B$.

The FN equation can be formulated as $\ln(I_D/V_D^2) = \alpha + \beta/V_D$, where $\alpha$ and $\beta$ have relevance to the charge density and tunneling energy barrier, respectively. Thus the linearity between $\ln(I_D/V_D^2)$ and $1/V_D$ confirms the FN tunneling[38]. By assuming that the graphene is a single emitter, $\alpha$ and $\beta$ were uniquely determined, as follows. The barrier height was obtained from the modified FE equation: $\ln(I/V^2) = \alpha + \beta/V$, where $\alpha$ and $\beta$ are $\ln\frac{A_{eff}q^3m}{8\pi h\Phi_B d^2 m^*}$ and $-\frac{8\pi\sqrt{2m^*}d}{3hq}\Phi_B^{\frac{3}{2}}$, respectively. First, the output characteristic of a FEB was measured for a FEB with a gate dielectric of 21.5 nm and a tunneling channel of 83.8 nm, as shown in Fig. 2a. Then a straight line of which slope is $\beta$ was obtained by replotting the output characteristic of a FEB according to the modified FE equation. $\beta$ includes a parameter of the FE barrier height. Therefore, $\beta$ declined with increasing $V_G$ and the FE barrier height decreased from 2.01 eV to 1.84 eV with increasing $V_G$ from 2 V to 8 V, as exhibited in Fig. 2b.

**Optimizing device performances.** Device characteristics—work function modulation of graphene, intrinsic gain, $I_{ON}$, delay ($\tau$), cut-off frequency ($f_T$), and power-delay product (PDP)—of the

semiconductor-less FEBs were investigated by varying $t_{Gate}$ and $t_{TC}$, where $\tau$ is a time delay required to charge gate electrode with $I_{ON}$, $f_T$ is a figure of merit of analog transistors in terms of switching speed, and PDP is that of digital ones in terms of required energy for switching[12,39,40]. The $t_{Gate}$ and $t_{TC}$ affect the amplitude of the graphene work function modulation, tunneling-barrier height, and thus device performances. First, the capacitive coupling governs how effectively the $V_G$ accumulates charges in the graphene as observed in GFET. The capacitive coupling or quantum capacitance ($C_Q$) of the graphene in the GFET has been determined to reduce the work function modulation because the $C_Q$ is serially connected to the gate capacitance $C_{Gate}$ (Fig. 3a) and, consequently, consumes a portion of $V_G$. Therefore, the accumulated charge reduces to $C_Q C_{Gate}/(C_Q + C_{Gate})$ multiplied by the $V_G$, where the larger the $C_{Gate}$, the higher the effect of $C_Q$, resulting in the smaller accumulated charge on the graphene[41,42]. However, the FEB involves a more complex network of capacitors because of the additional tunneling-channel capacitor ($C_{TC}$), as shown in Fig. 3b. As described in the supplementary text, the potential difference of the graphene from the Dirac point ($\varphi_{gr}$) in the FEB is determined by the following equation (Please see "Capacitive coupling among $C_{TC}$, $C_G$ and $C_Q$" section in the Supplementary Material.):

$$C_{Gate}V_G + C_{TC}V_D = \frac{e}{\pi}\left(\frac{e}{\hbar\nu_F}\right)^2 \varphi_{gr}^2 + (C_{Gate} + C_{TC})\varphi_{gr}. \quad (2)$$

The left side of the equation is the fictitious charge ($Q_{fic}$) on graphene accumulated by varying both the operating conditions ($V_G$ and $V_D$) and the device structures ($C_{Gate}$ and $C_{TC}$); the right

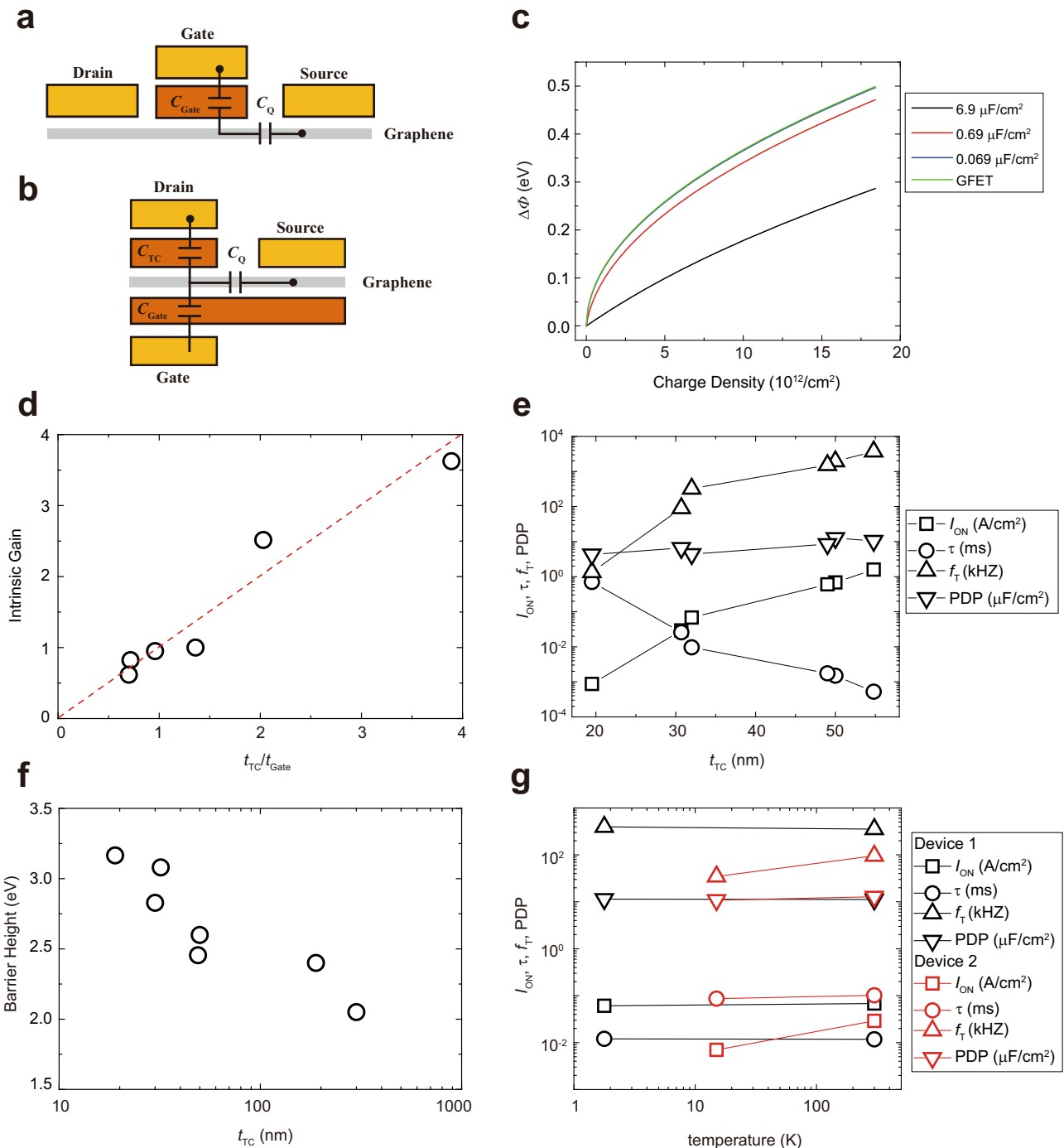

**Fig. 3 Capacitive coupling, intrinsic gain, and device performances.** Capacitive coupling of **a** GFET and **b** FEB. **c** Work function shift by varying the $C_{total} = C_{TC} + C_{Gate}$. $C_{total} = 6.9\,\mu F/cm^2$ when $t_{Gate} = 1\,nm$ and $t_{TC} = 1\,nm$; $C_{total} = 0.69\,\mu F/cm^2$ when $t_{Gate} = 10\,nm$ and $t_{TC} = 10\,nm$; and $C_{total} = 0.069\,\mu F/cm^2$ when $t_{Gate} = 100\,nm$ and $t_{TC} = 100\,nm$. The work function modulation of GFET is the upper limit of that of the FEB. **d** Intrinsic gain ($g_m/g_{ds}$) by varying the $t_{TC}/t_{Gate}$. The intrinsic gain is proportional to the $t_{TC}/t_{Gate}$ (the red dotted line is for guidance). **e** Device performances when $t_{TC}$ is 19.5, 30.7, 32, 49, 50, and 54.8 nm, and $t_{Gate}$ is 27.8, 42.8, 33, 36, 52, and 54.4 nm, respectively. $I_{ON}$, $1/\tau$, $f_T$, and PDP increase with $t_{TC}$. They increase to ~1000 times as $t_{TC}$ increases by ~35 nm, except for PDP. **f** Field-emission barrier height by varying $t_{TC}$, extracted by single-emitter approximation. The barrier height between graphene's Dirac point and the conduction band decreases as $t_{TC}$ increases. It decreases by 1.2 eV, as $t_{TC}$ increases from 19.5 to 301 nm. **g** Temperature-dependent performances of FEB. $I_{ON}$ of the most FEBs (e.g., device 1, black shapes) varies only 11.5% as temperature increases from 1.78 to 300 K; $\tau$ does <2.1%; $f_T$ does 10.6%; PDP does 1.5%. In contrast, some devices such as 2 (red shapes) exhibited temperature-dependent performances: $I_{ON}$ varies 314%; $\tau$ does 17.9%; $f_T$ does 177%; PDP does 17.9%.

side demonstrates how the $Q_{fic}$ determines $\varphi_{gr}$ with the coupling of $C_{Gate}$ and $C_{TC}$. Both the $C_{TC}$ and $C_{Gate}$ govern the work function shift in an identical manner (In conventional transistors, $C_{Gate}$ and body capacitance ($C_{Body}$) also exist in the Si

substrate. Their turn-on state was achieved when the minority charge accumulated on the channel, resulting in inversion. In the inversion state, $C_{Body}$ has no role in the capacitive coupling. Therefore, $C_{Gate}$ is the most critical capacitance, and we are less

concerned about capacitive coupling.). The work function shifts with the accumulated charge and can be obtained for the GFET and FEB by varying the $C_{total} = C_{TC} + C_{Gate}$, as shown in Fig. 3c. A smaller $C_{total}$ produces a larger work function modulations of graphene by the same amount of charges (x-axis). Therefore, the minimum value of the $C_{total}$ should be targeted to improve the $I_{ON}/I_{OFF}$, and the upper limit of the shift with a fixed $C_{Gate}$ can be determined when the $C_{TC}$ becomes 0 (i.e., the case of the GFET). The above coupling analysis is generally applicable to other vertical devices, including field-effect tunneling transistors, vertical field-effect transistors (vFETs), thin-film barristors, and GBs, that rely on the work function modulation of graphene, as shown in Supplementary Fig. 1. Furthermore, the modulation can be improved by engineering the capacitance—the dielectric constant and the thickness, as described in Supplementary Note 3.

Second, the intrinsic gain of FEB, obtained by the ratio of transconductance ($g_m$) to drain conductance ($g_{ds}$)[12], is proportional to the $C_{Gate}$–$C_{TC}$ ratio ($t_{TC}$–$t_{Gate}$ ratio), as shown in Fig. 3d. Since they have not reported the gain of vertically stacked devices[13], there is some doubt that the devices could not amplify (intrinsic gain <1). However, we obtained the gains of 2.5 and 3.6 using FEBs with the $C_{Gate}$–$C_{TC}$ ratios of 2.0 and 3.9, respectively. Along with the other 4 FEBs, we clarified that the intrinsic gain is proportional to the $C_{Gate}$–$C_{TC}$ ratio, as exhibited in Fig. 3d. It is because the fictitious charge, which determines the work function of the graphene, is linearly related to both $V_G$ and $V_D$ by a coupling between the $C_{Gate}$ and $C_{TC}$, described in the above equations.

Lastly, the other performances such as $I_{ON}$, $\tau$, $f_T$, and PDP, were governed by $t_{TC}$, as exhibited in Fig. 3e. As $t_{TC}$ increases from 19.5 to 54.8 nm, $I_{ON}$, $\tau$, and $f_T$ are dramatically improved (~1000 times): $I_{ON}$ increases from 0.87 mA/cm² to 1.59 A/cm²; the $\tau$ decreases from 0.7 ms to 0.52 μs; $f_T$ increases from 0.21 kHz to 0.59 MHz. It is because the $I_{ON}$ exponentially depends on the barrier height at the graphene–hBN junction, which decreases with $t_{TC}$, as shown in Fig. 3f. The energy difference between graphene's Dirac point and the conduction band of 19.5-nm thick hBN is obtained as 3.2 eV using single-emitter approximation; that of 301-nm thick one decreased to 2.1 eV. It is common for electron affinity of 2D materials to decrease with their thickness[9,43]. Therefore, they can improve the device performances by increasing the $t_{TC}$. However, the thicker the $t_{TC}$, the greater is the $V_D$ required for field emission from the graphene to the drain electrodes. It is why the PDP increases(or worsens) as the $t_{TC}$ increases: as the $t_{TC}$ increases from 19.5 to 54.8 nm, the PDP increases from 4.3 to 10.4 μJ/cm². Moreover, the lower the barrier height, the more dominant the temperature-dependent current. For example, vFETs with graphene–WS₂ heterostructure exhibited temperature-dependent performances: the $I_{ON}$ increased by around 1 order, and the $I_{ON}/I_{OFF}$ decreased by approximately 2 orders[19]. The dependence originates from the transport mechanism of the vFET: the thermionic emission. Therefore, when optimizing the barrier height of the semiconductor-less transistor, the upper limit is determined by the device performances—$I_{ON}$, $\tau$, and $f_T$—and the lower limit is by PDP and the thermionic emission current.

Notably, temperature independence of the performances is the unique property of the semiconductor-less vertical transistor with field-emission current, as shown in Figs. 1j and 3g. Most FEBs exhibited temperature-independent performances. A representative device's performances are shown in Fig. 3g (black shapes), where device parameters varied by only 1.5% or up to 11.5%. However, some devices such as device 2 (red shapes) exhibited a little more dependence on temperature (from 17.9 to 314%).

We understand that the temperature-dependent characteristics of the semiconductor-less devices originate from Poole–Frenkel transport mediated by intrinsic defects of hBN aggregated in its defect-rich domain[44]. The analysis is described in Supplementary Note 4.

Consequently, the result indicates that the device performances of FEB can be engineered in different ways as follows: (1) the switching of FEB is governed by the capacitive coupling. (2) The intrinsic gain is proportional to the $C_{Gate}$–$C_{TC}$ ratio. (3) The barrier height of graphene–hBN junction decreases with the $t_{TC}$. (4) The thicker the $t_{TC}$, the better is the performance of $I_{ON}$, $\tau$, and $f_T$. At the same time, PDP is degraded and temperature-dependent portion of the current increases to induce the temperature dependence of the FEB. Notably, all the characteristics of the semiconductor-less devices with optimized barrier height are temperature independent, unless the defect-rich domain of hBN was used[44]. Therefore, an optimized device geometry (e.g., $t_{TC}$) is indeed a key to realize the temperature-independent transistors with industry-applicable performances.

## Discussion

We report the semiconductor-less solid-state switching device with an $I_{ON}/I_{OFF}$ of 10⁶ in which a ballistic current can be effectively modulated by electric gating; thus the device exhibits not only adjustable gain but also unprecedented temperature-independent performances, such as $I_{ON}$, $\tau$, $f_T$, and PDP. Moreover, we clarified the role of capacitive coupling among the $C_{Gate}$, $C_{TC}$, and $C_Q$ for the modulation of the graphene work function in the vertical device geometry. In our modeling, the $C_{TC}$ is as essential as the $C_{Gate}$. The capacitive coupling is universal for all vertically stacked devices based on van der Waals heterostructures, which exploit the work function modulation of the graphene as their main switching mechanisms. Our FEB achieves industry-applicable device operations with current stability over a wide range of the temperature, which resolves the long-standing issue in conventional semiconductor-based transistors and extends the potential of 2D van der Waals devices to applications in extreme environments.

## Methods

**Device fabrication**. Monolayer graphene and two samples of hBN were prepared by mechanical exfoliation method. It was verified that the graphene is monolayer by using Raman spectroscopy, and the thickness of the hBN was measured by atomic force microscope. To make metal/hBN/graphene/hBN/metal vertical structure, the conventional wet transfer method and dry transfer method, which is called polydimethylsiloxane (PDMS) stamping, were conducted[45,46]. First, the hBN flakes were exfoliated onto PDMS surface to find several samples of few layer hBN. After finding two samples of few layer hBN on each PDMS surface using optical microscope, one was transferred onto exfoliated monolayer graphene on SiO₂ substrate by using the PDMS stamping method, and the other one was transferred onto Au/Cr gate electrode, which was deposited on 300 nm SiO₂ substrate. Second, the sample of hBN/graphene was coated with 950 K PMMA C4 at 4500 rpm by using spin coater. After that, the PMMA-coated hBN/graphene was transferred onto the hBN/metal structure by using the conventional wet transfer method. Third, in case that total thickness of the heterostructure was thicker than 80 nm, the metal/hBN/graphene/hBN junction was coated with the PMMA to make a PMMA bridge. The PMMA was cross-linked by exposure to an electron beam with a very high dose (15,000 μC/cm²), and the top electrodes were deposited along the cross-linked PMMA by using electron beam lithography and electron beam evaporator.

**I–V measurement**. Field-emission current of the device was measured in vacuum probe station and physical property measurement system at various temperatures with Keithley 4200.

## Data availability

The authors declare that the data supporting the findings of this study are available within the paper and its supplementary information files.

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

## Acknowledgements

H.-J.C. acknowledges support from Samsung Research Funding & Incubation Center for Future Technology (SRFC) (SRFC-MA1502-08) and National Research Foundation of Korea (NRF) grants funded by the Korea government (MSIT) (NRF-2020R1A2C1003398) and (MOE) (NRF-2018R1D1A1B07050452). H.Y. acknowledges support from National Research Foundation of Korea (NRF) under NRF-2020R1A2B5B02002548. B.H.P. acknowledges support from the National Research Foundation of Korea (NRF) grants funded by the Korea government (MSIP) (No. 2013R1A3A2042120). S.H.J. acknowledges support from National Research Foundation of Korea (NRF) under NRF-2018R1A2B6003937. S.W.L. acknowledges support from the Basic Science Research Program (NRF-2019R1A2C1085641) through the National Research Foundation of Korea (NRF) funded by the Korea government (MSIP). K.W. and T.T. acknowledge support from the Elemental Strategy Initiative conducted by the MEXT, Japan, Grant Number JPMXP0112101001, JSPS KAKENHI Grant Number JP20H00354, and the CREST (JPMJCR15F3), JST.

## Author contributions

H.-J.C. suggested the concept of semiconductor-less vertical transistor. J.-H.L. stacked the samples to make a graphene/hBN vertical transistor and measured transport characteristics of the device. J.-H.L. and D.H.S. calculated modulation of tunneling barrier height and device performance. N.B.J. and D.-H.P. prepared single-layer graphene and few-layer *h*BN by using Raman microscope and AFM. K.W. and T.T. synthesized *h*BN single crystals. E.K. contributed to an analysis of tunneling barrier height. J.-H.L., D.H.S., H.Y., and H.-J.C. wrote the manuscript. S.W.L., S.H.J., B.H.P., and Y.K. contributed to analyze the results of experiment. All authors contributed to revise the manuscript.

## Competing interests

The authors declare no competing interests.
