## [Peer Review File · Nature Communications]

REVIEWER COMMENTS

Reviewer #1 (Remarks to the Author):

The authors have successfully demonstrated a vertical field emission barristor, which can optimize the source-drain tunneling current by graphene quantum capacitance effect. The authors have claimed their vertical device is semiconductor-less, temperature independent compared with devices in history. Here are my comments:

1. Concerning the materials, for semiconductor-less devices, one may notice the device THETA (tunneling hot-electron-transfer amplifier) reported by Mead in 1960, using a MIMIM (M: metal, I: insulator) structure (Pro. IRE, 48, 359 (1960); J. Appl. Phys. 32, 646 (1961)). Other types of THETA with a MIMIM structure have also been studied.

Most recently, a device using a structure of Gr-insulator-Gr-insulator-Gr has been reported where the function of insulator is realized by BN or WSe₂ (IEEE EDL, 39, 634 (2018)). Thus a semiconductor-less device is not a pioneering contribution of this work.

2. Concerning the mechanism for temperature independence, the authors have correctly pointed out this is attributable to the nature of FE tunneling. However, they only show a current stability in low and room temperature (15-300 K) and claim resolving a long-standing issue for the first time.

In the device with a similar structure of Gr-insulator-Gr-insulator-Gr (IEEE EDL, 39, 634 (2018)), a junction current stability near room temperature is also shown and attributed to the FN tunneling. However, the stability is lost when temperature rises because of the thermionic emission.

In modern industry, a issue is the device performance at high temperature because of the power issue. What is the current stability at high temperature (300-400 K) for this device? What is the advantage compared with the above work?

3. Concerning the mechanism for current conduction, the authors have correctly pointed out their work is based on the graphene barristors. It can be seen as a tunnel junction edition of barristor instead of the Schottky junction (Science, 336, 1140 (2012)). In that case, what is the advantage using a tunnel junction compared with a Schottky junction theoretically?

4. Concerning the structure, recently graphene base transistors have been proposed (IEEE EDL, 33, 691 (2012); IEEE TED, 60, 4263 (2013)), and realized experimentally by structures of Semiconductor-insulator-Graphene-insulator-Metal ((Nano. Lett., 13, 1435 (2013); Nano. Lett. 13, 2370 (2013); et al.) or Semiconductor-Graphene-Semiconductor (Nature Comm., 10, 4873 (2019)). On/off current ratio of $10^{>5}$ has already been achieved for vertical graphene base transistors. These graphene devices using the similar vertical structures should be introduced.

5. Fig.1:

- 1) What is V_{bias} in Fig.1d and e? If it is the drain bias, it should include not only the voltage drop on the insulator, but also the voltage drop on graphene because of the quantum capacitance effect.
- 2) How about the leakage of the device. Please show ID when $V_g < 0$ in Fig. 1f.

6. Fig.2:

What is the difference between an ideal graphene emitter and an array in terms of device performance?

7. Fig.3:

- 1) What is "Charge Density" in Fig.3c? What is the relationship between C_{total} and I_{on}/I_{off} mathematically, if possible?
- 2) Please provide more details of measurements of gain, delay, cut-off frequency, et al. Are they

directly measured or calculated? How did the authors do the calculations and measurements?
What are the biases in the measurements?

3) Is there any direct experimental evidence for the barrier height variation with t_{TC} in Fig.3f? For example, experimental measurement results of the electron affinity?

8. SI

1) S1: Is the last equation correct?

2) S3: "...as shown in Fig. S3..." should be changed to "...as shown in Fig. S4...".

Reviewer #2 (Remarks to the Author):

A field emission process, also referred to as Fowler-Nordheim tunneling, is often discussed in the regime of high electrical field (for example, $E \sim 10^{10}$ V/m), especially in a vacuum condition.

In this present paper, the authors showed a device structure with a graphene/BN/Au tunnel junction (which is often seen in many conventional tunneling devices made out of graphene). The difference here is, the authors placed one more gate electrode in the bottom of the tunnel junction, separated by an additional BN dielectric. And the authors claim that their devices are able to work in the field emission regime.

By testing this above kind of device, the authors then drew their conclusions, mainly below:

1. While most solid state semiconducting devices work in the form of scattering-limited charge transport, which is reflected as a strong temperature dependent device performance, the gated vertical tunnel device (named as 'Field emission barristor', FEB, by the authors) show rather constant behavior, in a wide temperature range, from 15 K to 300 K.

2. The gain of the output characteristics can be adjustable via the bottom gate, with an I_{ON}/I_{OFF} reaching a value of 10 to the power of 6.

In general, I found this work of some interests, and I do appreciate the detailed analysis using SK-plots and the quantum capacitance model in the manuscript.

However, there are, at least in its present form, quite some flaws or unclear messages that have to be improved/clarified.

My major concerns are as follows (the manuscript file is not numbered at the bottom of each page, so, let's start the number from 1 of the title/abstract page):

1. Page 6, the authors cited Ref [28],[33] to specify that "the FE of planar graphene should be treated as an array of numerous emitters because of electron-hole puddles exist in graphene". However, the so-called electron-hole puddles picture is only valid at the charge neutrality point (Dirac point) of graphene. The authors' data are apparently in the electron doped regime and the electron-hole puddle image should not be true.

2. A very straightforward question is: What makes the big difference between the currently studied devices and those studied in Britnell et al's work (Ref. [25] in this manuscript)? I mean, why the authors see the crossover from direct tunneling (DT) to the field emission (FE), but Britnell et al only saw DT?

3. The whole paper is defining the structure as a "semiconductor-less" transistor, yet there is still

h-BN used as the tunnel barrier. But, h-BN is definitely a semiconductor, no? It is well known that the bulk h-BN is wide-gap semiconductor, with an indirect band gap of ~ 6 eV. I don't understand why the authors name this as "semiconductor-less" transistor. It should rather be defined as a graphene/BN/Au tunneling junction equipped with an additional bottom gate, separated by another h-BN dielectric.

4. Why the authors did not see the Dirac point in their devices? For example, as shown in Fig. 1f, there is no hole-side seen in the field effect curve.

Some additional comments:

1. Page 2, the authors claim that "the enhanced electric field degrades the carrier mobility in the semiconductor channel", why? Can the authors put a bit more discussions to explain?

2. Page 2, "have attracted enormous interest because they utilize ballistic ..., However, industry-applicable current switching was not realized in the tunneling devices." Here, this sentence might not be true, and I would suggest the authors to cite the recent paper from NASA, Nature Electronics 2, 405–411(2019), in which Han et al realized wafer-scale vacuum channel transistors.

3. Page 3, "The ideal solution ...only one-order modulation of I_{ON}/I_{OFF} has been reported.[25]" This sentence is not proper. In reference [25], Britnell et al did not use any gate electrodes in their structure. And we usually do not define the non-linear output curve (IV curve) as an I_{ON}/I_{OFF} ratio. The term I_{ON}/I_{OFF} is used to describe a transfer curve (or a field effect curve).

4. It is seen that the largest V_G in Fig. 1f is about 20 V; but the largest V_G in Fig. 1g-1h is cut off at 12 V. As a reader, I'm curious to see how does the IV look like at higher V_G ? And how do the IV curves look like when the source-drain is negatively biased (negative V_D)? Are the IV curves symmetric with respect to positive and negative V_{DS} ?

5. Page 5, "The FE current I_D , which increases..., can be described as follows:", please cite the original paper when writing that formula.

Reviewer #1 (Remarks to the Author):

The authors have successfully demonstrated a vertical field emission barristor, which can optimize the source-drain tunneling current by graphene quantum capacitance effect. The authors have claimed their vertical device is semiconductor-less, temperature independent compared with devices in history. Here are my comments:

1. Concerning the materials, for semiconductor-less devices, one may notice the device THETA (tunneling hot-electron-transfer amplifier) reported by Mead in 1960, using a MIMIM (M: metal, I: insulator) structure (Pro. IRE, 48, 359 (1960); J. Appl. Phys. 32, 646 (1961)). Other types of THETA with a MIMIM structure have also been studied.

Most recently, a device using a structure of Gr-insulator-Gr-insulator-Gr has been reported where the function of insulator is realized by BN or WSe₂ (IEEE EDL, 39, 634 (2018)). Thus a semiconductor-less device is not a pioneering contribution of this work.

We appreciate the reviewer's comment on the comparison between former 'semiconductor-less devices' and our field emission barristor (FEB). It should be noted that, as written in the title of our manuscript, we aimed 'semiconductor-less transistor with a high switching ratio up to $I_{ON}/I_{OFF}=10^6$ ' rather than 'any devices (e.g., bipolar junction transistors, BJTs) without semiconductors'.

The devices mentioned by reviewer #1 are BJT-type devices, where the main figure of merit (FOM) is a common-base current gain (I_C/I_E). However, in our barristor that operates as a field-effect transistor (FET), the critical FOM is I_{ON}/I_{OFF} .

For example, the 2D-HET reported in IEEE EDL, 39, 634 (2018) exhibits the gain of 99.95% without analyzing its switching. While the switching is not a FOM of the 2D-HET, we compared

the $I_{\text{ON}}/I_{\text{OFF}}$ of the 2D-HET to that of the FEB, as follows. After extracting the modulation of I_C by varying V_{BE} with constant V_{CB} , we analyzed the transconductance from Fig. 3 (c) in the reference: $I_C^{\text{ON}}/I_C^{\text{OFF}} < 2$ under operating condition and $I_C^{\text{ON}}/I_C^{\text{OFF}} < 6$ under the whole applied voltage range. **Therefore, the BJTs in previous reports without semiconductors should not degrade the novelty of our work, ‘semiconductor-less transistor with a high switching ratio up to $I_{\text{ON}}/I_{\text{OFF}}=10^6$ ’.**

2. Concerning the mechanism for temperature independence, the authors have correctly pointed out this is attributable to the nature of FE tunneling. However, they only show a current stability in low and room temperature (15-300 K) and claim resolving a long-standing issue for the first time.

In the device with a similar structure of Gr-insulator-Gr-insulator-Gr (IEEE EDL, 39, 634 (2018)), a junction current stability near room temperature is also shown and attributed to the FN tunneling. However, the stability is lost when temperature rises because of the thermionic emission.

In modern industry, a issue is the device performance at high temperature because of the power issue. What is the current stability at high temperature (300-400 K) for this device? What is the advantage compared with the above work?

We agree on the reviewer’s comment: the importance of the device performance at 400 K. To respond to the reviewer’s comment, we conducted I - V measurement with the temperature range 15-400 K, to improve the manuscript. The Fig. 1j in the revised manuscript exhibits temperature-independent switching performance of our FEB when negative (red, p-type) and positive (black, n-type) drain voltage were applied. At 400 K, the switching performance is not degraded for the following reason. Since the tunneling barrier height between the graphene and the $h\text{BN}$ is estimated to be 2.1 eV for electron, thermally excited charge carriers in graphene could not escape over the barrier high. In contrast, for the device of Gr-insulator-Gr-insulator-Gr (IEEE EDL, 39, 634 (2018)), the thermal electron in graphene could escape over the graphene- WSe_2 barrier at 400 K because the barrier height is 0.6 eV and it is reduced to below 0.1 eV by the base voltage. Consequently, the advantage of our FEB is the temperature-independent switching behavior at high temperature due to the sufficiently large tunneling barrier height compared to the thermal energy. The revised text and Fig. 1j are shown below.

(the Fig. 1j in page 20)

(the Fig. 1j caption in page 21)

<< j , I_D switching under $V_D = -18$ V (red) and 29 V (black) also exhibited little temperature degradation from 400 K to 15 K.>>

3. Concerning the mechanism for current conduction, the authors have correctly pointed out their work is based on the graphene barristors. It can be seen as a tunnel junction edition of barristor instead of the Schottky junction (Science, 336, 1140 (2012)). In that case, what is the advantage using a tunnel junction compared with a Schottky junction theoretically?

This is an inspiring comment to make our work distinguished from conventional graphene barristor. In this study, we aimed temperature-independent barristor (excluding semiconductor components), maintaining the advantages of graphene such as high mobility and mechanical flexibility for flexible devices.

Compared to the Schottky-junction current (i.e., thermionic emission), the field-emission current in our FEB is independent on the temperature, as shown in the following equations:

$$\text{(Field-emission current)} \quad J(V) = \frac{q^3 m V^2}{8\pi h \Phi_B d^2 m^*} \exp\left[-\frac{8\pi\sqrt{2m^*}\Phi_B^{\frac{3}{2}}d}{3hqV}\right] \quad (\text{Ref. 35}),$$

$$\text{(Schottky-junction current)} \quad J(V) = \frac{8\pi^2 q (k_B T)^3}{h^3 v_f^2} \exp\left(-\frac{\Phi_B - E_F(V_G)}{k_B T}\right) \left[\exp\left(\frac{qV}{nk_B T}\right) - 1\right] \quad (\text{Ref. 36}),$$

where q is the elementary charge, m is the electron mass, m^* is the electron tunneling mass, V is the applied voltage, h is Plank's constant, d is the tunneling distance, T is the temperature, k_B is the

Boltzmann constant, v_F is the Fermi velocity, $E_F(V_G)$ is Fermi-level difference of graphene from Dirac point with a gate voltage of V_G , Φ_B is the Schottky-barrier height with the graphene's Fermi-level at Dirac point, and n is the ideality factor of the junction.

Accordingly, our FEB has no semiconducting components and **realizes temperature-independent operations, which could be considered the advantage of our FEB theoretically**. To contrast the feature, we have added a sentence in the Result section of the manuscript as follows:

(the revised Results section in page 5)

<<... and d is the tunneling distance. While the Schottky current depends on the temperature, the FE current barely depends on the temperature³⁶. The above formula supports the critical device operation, the switching with an I_{ON}/I_{OFF} ratio of $\sim 10^6$, presented in Fig. 1g. ...>>

4. Concerning the structure, recently graphene base transistors have been proposed (IEEE EDL, 33, 691 (2012); IEEE TED, 60, 4263 (2013)), and realized experimentally by structures of Semiconductor-insulator-Graphene-insulator-Metal ((Nano. Lett., 13, 1435 (2013); Nano. Lett. 13, 2370 (2013); et al.) or Semiconductor-Graphene-Semiconductor (Nature Comm., 10, 4873 (2019)). On/off current ratio of 10^5 has already been achieved for vertical graphene base transistors. These graphene devices using the similar vertical structures should be introduced.

This comment would make our manuscript more complete to introduce graphene-based transistors. **We have added the references of BJT-like devices (the reviewer mentioned)**, where the thin metal base in BJT was replaced by graphene. As replied to the comment 1, **we aimed 'semiconductor-less transistor with a high switching ratio up to $I_{ON}/I_{OFF}=10^6$ ', which could not be realized in former BJTs with graphene;** I_{ON}/I_{OFF} is not a FOM for BJTs.

As the reviewer commented, vertical graphene transistors with semiconductor junctions have achieved switching operations up to a ratio of 10^5 ; Nano Lett. 13, 1435 and IEEE TED 60, 4263 achieved a ratio of 10^4 ; Nano Lett. 13, 2370 and Nature Comm. 10, 4873 achieved a ratio larger than 10^5 . However, **all the above devices for large I_{ON}/I_{OFF} require junctions with semiconductors, which underlines the novelty of our work**. As suggested by the reviewer, we have properly added and adjusted the references like below.

(the revised introduction in page 3)

<<... This principle was originally demonstrated in graphene barristors (GBs)¹⁴ and has been used in various devices containing either organic¹⁵⁻¹⁸ or inorganic¹⁹⁻²⁵ semiconductor-graphene junctions. In addition, bipolar junction transistor (BJT)-like devices have been also investigated, where graphene was used as a base material, thus called as graphene-base transistor²⁶⁻³⁰. The switching in such devices does not rely on the thermally generated charge of semiconductors, but semiconductors are still crucial elements required to achieve efficient switching. ...>>

5. Fig.1:

1) What is V_{bias} in Fig.1d and e? If it is the drain bias, it should include not only the voltage drop on the insulator, but also the voltage drop on graphene because of the quantum capacitance effect.

As the reviewer correctly pointed out, V_{bias} is the drain bias. The drain bias should involve the charge accumulation by the quantum capacitance.

We show a schematic picture for the voltage drop in Fig. R1 below. The potential drop by the quantum capacitance of graphene depends on the amount of charges accumulated in the graphene, resulting in modulation of the work function of the graphene.

Figure R1. Voltage drop with and without work function modulation (quantum capacitance) in graphene.

Given that the dielectric (h BN tunneling barrier) is thick, the voltage drop by the quantum capacitance of graphene is negligible compared to the voltage drop across the dielectric. For example, the work function modulation in a FEB is smaller than 0.3 eV with a drain voltage of 20 V; the voltage drop by the quantum capacitance is only 1.5% of that occurring in the dielectric. Therefore, we did not seriously consider the quantum capacitance in our manuscript. Nevertheless, the equation in S1 includes the C_Q to rigorously describe the effect of the work function modulation.

2) How about the leakage of the device. Please show ID when $V_g < 0$ in Fig. 1f.

As suggested by the reviewer, we have added the gate leakage current in the revised Fig. 1f (Fig. 1g in the revised manuscript). When $V_G < 0$ V, the FEB is in OFF state. Although the gate leakage current increases dramatically with $V_G > 14$ V, the leakage current remains less than 0.5 % of the drain current in the range. The Fig. 1g and revised text are shown below.

(the Fig. 1g in page 20)

(the Fig. 1g caption in page 21)

<<... The gate voltage increases from Fig. c to f. **g**, I_D switching performance of the semiconductor-less device. I_{D-} and I_{G-} are drain and gate current under $V_D = -18$ V. I_{D+} and I_{G+} are drain and gate current under $V_D = 29$ V. For the negative V_G , I_{ON}/I_{OFF} above 10^6 has been achieved at 300 K. **h**, Device characteristics ...>>

(the revised Result section in page 5)

<<... **Transport characteristics of semiconductor-less transistor.** In Fig. 1g, the FEB with a gate-*h*BN thickness (t_{Gate}) of 62 nm and a tunneling-*h*BN thickness (t_{TC}) of 64 nm shows an efficient switching (by an $I_{\text{ON}}/I_{\text{OFF}}$ of up to 10^6) without semiconductors. ...>>

(the revised Results section in page 5)

<<... Indeed, former graphene-based tunneling or lateral devices based on DOS-dependent channel current have $I_{\text{ON}}/I_{\text{OFF}}$ ratios of ~ 10 , which is a physical limit imposed by the fact that the charge density modulation is limited to 100 at room temperature³⁷. **As V_G increases, the tunneling mechanism of the electrons for the gate leakage current is changed from DT to FE at $V_G = 14$ V or a gate field of 0.23 V/nm in a similar manner with I_D . The I_G remains less than 0.5 % of I_D in the V_G range. Minimizing the leakage effect on the I_D , the gate field was limited to 0.23 V/nm in our study.**>>

6. Fig.2:

What is the difference between an ideal graphene emitter and an array in terms of device performance?

We appreciate the comment on the realistic system. In terms of device performance, we expect the same performances from an ideal graphene emitter and an array of emitters. The reason is that **the charge puddles (treated as an array of emitters in our manuscript) in graphene become dominant only when the Fermi level of graphene is located at the Dirac point.** In the operation of our FEB, the Fermi level is located fairly above the Dirac point (~ 0.3 eV), so the effect of arrays is negligible.

In the original manuscript, we wanted to simulate the most realistic case. In our revised one, we modified the analysis from an array to an ideal emitter (Fig. 2).

7. Fig.3:

1) What is “Charge Density” in Fig.3c? What is the relationship between C_{total} and I_{ON}/I_{OFF} mathematically, if possible?

The ‘Charge Density’ in Fig. 3c (x-axis) is the accumulated charge density in graphene by a Fermi level shift with respect to the Dirac point (y-axis). The green curve in Fig. 3c follows the well-known equation described in Supplementary Information S1 (also shown below).

The relationship between C_{total} and I_{ON}/I_{OFF} is not mathematically simple, so we did not present this in the manuscript. But, as the reviewer requested, we can explain the relationship in this response letter like below.

To derive I_{ON}/I_{OFF} as a function of C_{total} , we start from the equation in S1:

$$C_{Gate}V_G + C_{TC}V_D = \frac{e}{\pi} \left(\frac{e}{\hbar v_F} \right)^2 \varphi_{gr}^2 + (C_{TC} + C_{Gate})\varphi_{gr}.$$

Then, the work function modulation is obtained like below:

$$\varphi_{gr} = -\frac{C_{TC}+C_{Gate}}{2\frac{e}{\pi}\left(\frac{e}{\hbar v_F}\right)^2} \pm \sqrt{\left(\frac{C_{TC}+C_{Gate}}{2\frac{e}{\pi}\left(\frac{e}{\hbar v_F}\right)^2}\right)^2 + \frac{C_{Gate}V_G+C_{TC}V_D}{\frac{e}{\pi}\left(\frac{e}{\hbar v_F}\right)^2}}.$$

Since φ_{gr} is the deviation of Fermi level from the Dirac point with $V_G = V_D = 0$, the I_{ON}/I_{OFF} could be written as a function of φ_{gr} or C_{total} like below:

$$\frac{I_{ON}}{I_{OFF}} = \frac{\Phi_{B,OFF}}{\Phi_{B,ON}} \exp \left[\frac{-8\pi\sqrt{2m^*}d}{3hqV_D} (\Phi_{B,ON}^{3/2} - \Phi_{B,OFF}^{3/2}) \right]$$

where $\Phi_B = \chi_{Semiconductor} - \varphi_{gr} + 4.5$.

2) Please provide more details of measurements of gain, delay, cut-off frequency, et al. Are they directly measured or calculated? How did the authors do the calculations and measurements? What are the biases in the measurements?

Our analysis to calculate those parameters is primarily based on the equations in Ref 11, 38, and 39 in the original manuscript.

Firstly, the intrinsic gain, G , is defined as

$$G = \frac{g_m}{g_{ds}}$$

where transconductance $g_m = dI_D / dV_{GS}$ and drain conductance $g_{ds} = dI_D / dV_{DS}$. We measured various I - V characteristics of FEBs with different t_{TC} / t_{Gate} , and calculated the intrinsic gain based on the g_m and the g_{ds} extracted from I_D - V_{GS} curve and I_D - V_{DS} curve, respectively (following the above equation).

Secondly, the delay time, τ , is described below:

$$\tau = \frac{Q_{on} - Q_{off}}{I_{on}}$$

where I_{on} is current in the ON state, Q_{on} and Q_{off} indicate the accumulated charges on graphene by the gate and drain biases at ON- and OFF-states, respectively. In the schematic diagram of our FEB (Fig. 3b), the accumulated charges on graphene, Q , could be defined as $Q = C_{TC}V_{ds} + C_{Gate}V_{GS}$. So, when the V_{ds} is fixed, Q_{on} - Q_{off} could be expressed as follows:

$$Q_{on} - Q_{off} = C_{Gate}V_{ON} - C_{Gate}V_{OFF},$$

where V_{ON} is gate bias in the ON state, V_{OFF} is gate bias in OFF state. Thus, we measured I_D - V_{GS} characteristics of the devices with different t_{TC} , and extracted the Q_{on} , Q_{off} , I_{on} from the I_D - V_{GS} curve. C_{Gate} was calculated by taking the dielectric constant of hBN (3.9). From these values, we could calculate the delay time (τ) for the devices.

Thirdly, the power-delay product (PDP) is defined as

$$PDP = V_{DD} (Q_{on} - Q_{off}),$$

where V_{DD} is drain bias, and Q_{on} and Q_{off} indicate the accumulated charges on graphene at ON- and OFF-states, respectively. We calculated the PDP of the devices by extracting Q_{on} - Q_{off} from the I_D - V_{GS} curve.

Lastly, the cut-off frequency (f_T) of the device in ON-state is defined as

$$f_T = \frac{1}{2\pi} \frac{dI_D/dV_{GS}}{dQ/dV_{GS}},$$

where I_D is drain current, V_{GS} is gate bias, Q is accumulated charges on graphene by V_{GS} . Because dI_D/dV_{GS} can be extracted from I_D - V_{GS} curve and dQ/dV_{GS} can be calculated from the relation $Q = C_{Gate}V_{GS}$, we could calculate the cut-off frequency (f_T) of the devices with different t_{TC} .

3) Is there any direct experimental evidence for the barrier height variation with t_{TC} in Fig.3f? For example, experimental measurement results of the electron affinity?

We appreciate the critical comment from the reviewer. Indeed, the barrier height variation is key

for the operation of our field emission barristor (FEB). In our study, we used formula for field emission current, $I(V) = \frac{A_{\text{eff}}q^3mV^2}{8\pi h\Phi_B d^2m^*} \exp\left[-\frac{8\pi\sqrt{2m^*}\Phi_B^{\frac{3}{2}}d}{3hqV}\right]$, to estimate the barrier height, which could be considered as direct experimental evidence (by transport).

We note that **the barrier height in Fig. 3f was estimated with different electric fields (' d/V ' in the above formula) for FEB devices with different t_{TC} values**; to avoid possible noises, the electric field was chosen by the value to maximize the field emission current (just before the soft breakdown of the dielectric (*h*BN)).

Although the reviewer mentioned a possible change of the electron affinity, it is hard to consider such phenomenon in *h*BN with thicknesses ranging 20~300 nm (in FEB). Thus, we mainly investigated two possibilities for the barrier height variation: **1) the quantum capacitance of graphene under different electric fields (' d/V ') and 2) thickness-dependent work function (or surface potential) changes**. The different electric fields (' d/V '), used for the barrier height measurement, induce different amounts of charge accumulation in the graphene, which modifies the barrier height by up to 0.2 eV (the effect of the quantum capacitance of graphene). To further explain the change of barrier height of up to 1 eV in Fig. 3f, as requested by the reviewer, we need to clarify the other origin.

Figure R2. KPFM of *h*-BN layers on SiO₂/Si wafer. (a) Topography. (b) Surface potential (V_{surf}) of 20 nm and 200 nm thick *h*-BN layers on SiO₂/Si wafer. (c) Surface potential variation by the *h*-BN thickness (on SiO₂)

Figure R3. KPFM of *h*-BN layers on HOPG. (a) Topography. (b) Surface potential (V_{surf}) of 20 nm and 200 nm thick *h*-BN layers on HOPG. (c) Surface potential variation by the *h*-BN thickness (on graphite)

We conceived Kelvin probe force microscopy (KPFM) to newly investigate the thickness-dependent band alignment in our FEB. For this purpose, two systems were chosen and compared: *h*BN layers with various thicknesses located on SiO₂ (Fig. R2) and on graphite (HOPG) (Fig. R3) where surface potential mapping and thickness-dependent behaviors were measured. As shown in Fig. R2 (c) below, the surface potential of *h*BN/SiO₂ does not change by the thickness of the *h*BN: no charge exchange between *h*BN and the insulating SiO₂.

On the other hand, the (top) surface potential of *h*BN/graphite is modulated by the thickness of the *h*BN as shown in Fig. R3 (c), which indicates a charge exchange between the *h*BN and graphite. The defect states (also used to explain Poole-Frenkel transport in Fig. S5) might play a role for the charge exchanges in the *h*BN.

Considering the thickness-dependent surface potential (V_{surf}) and the different electric fields (the slopes of *h*BN for the determination of the barrier height in transport), two cases of band alignments could be drawn in Fig. R4. **In the band diagram (obtained by KPFM), the barrier height decrease is found to be ~ 1.05 eV, which matches the transport in Fig. 3f.** Accordingly, the threshold electric field (before the soft breakdown) modulates the barrier height in FEB involving *h*BN with different thicknesses.

Figure R4. Energy band diagram between the metal tip and *h*-BN on HOPG in KPFM. The V_{surf} is determined by the work function difference (vacuum level difference) between the tip and sample. E_{vac} , E_C , E_v , E_F are the vacuum level, the conduction and valence band edges, and the Fermi energy level, respectively.

8. S1

1) S1: Is the last equation correct?

We appreciate the keen observation of the reviewer. We corrected the equation as follows.

(the revised S1 in page 2)

<<... Considering the small modulation of drain and gate voltages with ϕ_{gr} (or I_D) unchanged, the above equation can be

$$C_{Gate}dV_G + C_{TC}dV_D = 0.$$

Therefore, the gain of V_D over V_G can be estimated with the ratio of C_{Gate}/C_{TC}>>

2) S3: "...as shown in Fig. S3..." should be changed to "...as shown in Fig. S4...".

We appreciate the comment. We corrected the sentence as the reviewer mentioned.

(the revised S3 in page 4)

<<... we calculated the graphene work function shifts by varying the high-k material and its thickness for C_{Gate} , as shown in Fig. S4. The upper left part of the graph shows the work function shift with an HfO_x gate dielectric; ...>>

Reviewer #2 (Remarks to the Author):

A field emission process, also referred to as Fowler-Nordheim tunneling, is often discussed in the regime of high electrical field (for example, $E \sim 10^{10}$ V/m), especially in a vacuum condition.

In this present paper, the authors showed a device structure with a graphene/BN/Au tunnel junction (which is often seen in many conventional tunneling devices made out of graphene). The difference here is, the authors placed one more gate electrode in the bottom of the tunnel junction, separated by an additional BN dielectric. And the authors claim that their devices are able to work in the field emission regime.

By testing this above kind of device, the authors then drew their conclusions, mainly below:

1. While most solid state semiconducting devices work in the form of scattering-limited charge transport, which is reflected as a strong temperature dependent device performance, the gated vertical tunnel device (named as 'Field emission barristor', FEB, by the authors) show rather constant behavior, in a wide temperature range, from 15 K to 300 K.
2. The gain of the output characteristics can be adjustable via the bottom gate, with an I_{ON}/I_{OFF} reaching a value of 10 to the power of 6.

In general, I found this work of some interests, and I do appreciate the detailed analysis using SK-plots and the quantum capacitance model in the manuscript.

However, there are, at least in its present form, quite some flaws or unclear messages that have to be improved/clarified.

My major concerns are as follows (the manuscript file is not numbered at the bottom of each page, so, let's start the number from 1 of the title/abstract page):

We appreciate the encouraging comments from Reviewer #2. As the reviewer mentioned, we tried our best to improve our manuscript below. The page numbers have also been added in the revised manuscript.

1. Page 6, the authors cited Ref [28],[33] to specify that "the FE of planar graphene should be treated as an array of numerous emitters because of electron-hole puddles exist in graphene". However, the so-called electron-hole puddles picture is only valid at the charge neutrality point (Dirac point) of

graphene. The authors' data are apparently in the electron doped regime and the electron-hole puddle image should not be true.

The reviewer correctly pointed out the effect of an array of emitters in FEB. We agree with the reviewer that electron-hole puddles are valid only when the Fermi-level is around Dirac point. We intended to provide a more general picture for the field emission from planar graphene in the original manuscript.

Following the reviewer's suggestion, the single emitter approximation in the Supplementary Information (with Fig. S2) has been moved to the Result section, and the array approximation (with Fig. 2) has been moved to the Supplementary Information with minor corrections: Φ_B for electron. The revised parts are like below.

(the revised Results section in page 6)

<<... By assuming that the graphene is a single emitter, α and β were uniquely determined, as follows. The barrier height was obtained from the modified FE equation: $\ln(I/V^2) = \alpha + \beta/V$, where α and β are $\ln \frac{A_{eff} q^3 m}{8\pi h \Phi_B d^2 m^*}$ and $-\frac{8\pi\sqrt{2m^*}d}{3hq} \Phi_B^{\frac{3}{2}}$, respectively. First, the output characteristic of a FEB was measured for a FEB with a gate dielectric of 21.5 nm and a tunneling channel of 83.8 nm, as shown in Fig. 2a. Then, a straight line of which slope is β was obtained by replotting the output characteristic of a FEB according to the modified FE equation. β includes a parameter of the FE barrier height. Therefore, β declined with increasing V_G , and the FE barrier height decreased from 2.01 eV to 1.84 eV with increasing V_G from 2 V to 8 V, as exhibited in Fig. 2b.>>

(the revised Supporting S2 in page 3)

<<S2. Array approximation of field emission from graphene near Dirac point.

As shown in Fig. S2a, the FE of planar graphene could be treated as an array of numerous (FE current) emitters because Electrical inhomogeneity exists in graphene^{1,2}. The FE current was analyzed based on such an array using Seppen-Katamuki (SK, or "intercept-slope" in Japanese) plots³ by extracting the slope and y-intercept values from the drain current using the axes $\ln(I_D/V_D^2)$ and $1/V_D$. For this analysis, the FE currents of the device measured at 15 K were plotted in Fig. S2b. Then, the SK-plots were obtained, as described in Fig. S2c.

Unlike the single-emitter assumption, which appears as a single point in the SK-plot, the linear distribution in Fig. S2c implies the log-normal distributions of the radii or heights of the emitters. The absolute value of the slope in the SK-plot decreases as the gate bias increases. Since the slope in the SK-plot is proportional to $\Phi_B^{3/2}$ (ref. 5), Φ_B for electron decreases by 8 % as the gate bias increases from 0 V to 12 V. Although the Fermi level of graphene varies according to the presence of impurities and defects⁶, we can assume that the Fermi level of our device is at the Dirac point when $V_D = V_G = 0$ V because of the *h*BN-graphene-*h*BN sandwich structure. Then, we obtained that the slope of the SK-plot raised to the power of 2/3 monotonically decreases relative to the charge carrier density raised to the power of 1/2 (Fig. S2d). The linear relationship between the slope in the SK-plot and the accumulated charges in the graphene validates the analysis because both are proportional to the work function of graphene.>>

(the revised Supporting Fig. S2 caption in page 7)

<<... The output characteristic of the FEB composed of monolayer graphene, gate-*h*BN (thickness : 42.8 nm), and tunneling-*h*BN (thickness : 30.7 nm). This graph exhibits a...>>

2. A very straightforward question is: What makes the big difference between the currently studied devices and those studied in Britnell et al's work (Ref. [25] in this manuscript)? I mean, why the authors see the crossover from direct tunneling (DT) to the field emission (FE), but Britnell et al only saw DT?

This is a critical comment to discuss the novelty of our work. Speaking from the conclusion, the key difference is the thickness of the tunneling barrier (*h*BN). Britnell *et al.* used thin ones (4 ~ 7 layers) with small biases (~0.2 V), and thus, observed DT with small electric fields (<0.04 V/nm). In our study, however, thick *h*BN layers (20 ~ 300 nm thick) were used with large electric fields (>0.35 V/nm); the thick tunneling barrier suppresses DT and makes the FE dominate the transport over DT.

To observe the FE, a high electric field (>0.35 V/nm) is required to make the band diagram, as shown in Fig. R5. In the study of Britnell *et al.*, increasing the electric field across the thin dielectric would cause a breakdown of the dielectric before the FE appears. This is the reason why

Britnell et al. could not observe the FE in their study.

Figure R5. Band diagrams for DT and FE. We need a high electric field across a thick dielectric layer.

3. The whole paper is defining the structure as a "semiconductor-less" transistor, yet there is still hBN used as the tunnel barrier. But, hBN is definitely a semiconductor, no? It is well known that the bulk hBN is wide-gap semiconductor, with an indirect band gap of ~ 6 eV. I don't understand why the authors name this as "semiconductor-less" transistor. It should rather be defined as a graphene/BN/Au tunneling junction equipped with an additional bottom gate, separated by another hBN dielectric.

In solid-state physics, both semiconductors and insulators have either fully-filled or empty bands, which is contrasted with metals having partially-filled bands. In the context, semiconductors and insulators are not clearly and easily separated.

However, we practically distinguish semiconductors and insulators by whether room-temperature energy ($k_B T \sim 25$ meV) (or a temperature variation around the energy) could affect the transport (e.g., resistivity) of the materials or not. Although some researchers may categorize hBN as a semiconductor, the resistivity of hBN is extremely high ($\sim 10^{14}$ $\Omega \cdot \text{cm}$) and cannot be modulated by practical temperatures (thermal energies) or gating. The resistivity features allow us to consider hBN as an insulator in our work. The keyword represented by 'semiconductor-less transistor'

indicates control of FE current (with a sufficient ON/OFF ratio) for the first time, which is distinguished from diffusive currents in ordinary semiconductor devices.

4. Why the authors did not see the Dirac point in their devices? For example, as shown in Fig. 1f, there is no hole-side seen in the field effect curve.

This comment highlights the originality of our work. Graphene-based FET (GFET) shows ambipolar behaviors due to the absence of the bandgap in graphene. Both electrons and holes flow as drain currents without OFF-state in graphene. However, **our FEB does not demonstrate the ambipolar behaviors, because the current is determined by the tunneling barrier height (rather than the density of states in graphene) which monotonically decreases with the charge, either electron or hole, accumulated on graphene.**

Instead, the work function modulation by quantum capacitance in graphene, thus barrier-height modulation, is enhanced around the Dirac point. It could be explained by the fact that the Fermi level modulation is proportional to the square root of the number of electrons or holes in graphene. Therefore, we do not observe the ambipolar behavior (seeing the Dirac point) in our FEB.

Some additional comments:

1. Page 2, the authors claim that "the enhanced electric field degrades the carrier mobility in the semiconductor channel", why? Can the authors put a bit more discussions to explain?

According to Ref. 1, at low electric fields, the velocity of the electrons in graphene is low, and the electrons scatter mostly with acoustic phonons; so, the velocity increases linearly proportional to the electric field. At high electric fields, the energy of electrons becomes high enough to scatter with optical phonons, saturating to a certain value: for example, the velocity saturates to 10^7 cm/s in silicon. To make the point clearer in the introduction, we have revised the abstract as follows:

(the revised introduction in page 2)

<<... in terms of increasing further spatial resolution of the device and temperature-dependent device performances in various environments. **The enhanced electric field degrades the carrier mobility in the semiconductor channel. It is because, with the enhanced electric field, the carrier starts to scatter with optical phonon of the semiconductors and lose more of its energy, resulting in**

the velocity saturation¹. Also, the carrier density or device performance depends on the temperature and, as a result, deviates from Moore's law^{2,3}. ...>>

2. Page 2, "have attracted enormous interest because they utilize ballistic ..., However, industry-applicable current switching was not realized in the tunneling devices." Here, this sentence might not be true, and I would suggest the authors to cite the recent paper from NASA, *Nature Electronics* 2, 405–411(2019), in which Han et al realized wafer-scale vacuum channel transistors.

We thank the reviewer for introducing the reference.

Han et al. studied vFETs in the 150-mm wafer-scale and achieved long-term stability of the device on SiC wafers. However, the reference demonstrated $I_{ON}/I_{OFF} \sim 126$, when V_d was 20 V, and V_g changed from 0 V to 20 V (left column in the 3rd page of the reference), which is distinguished from our switching operation. We note that, as stated in Table 2 in Ref. 39 (*Nature* **479**, 388-344 (2011)), the industrial requirement of the switching is $10^4 \sim 10^6$; limited transistor applications require $I_{ON}/I_{OFF} \sim 10^3$ with high ON-currents.

Still authors appreciate the above achievement. So, we have added the reference in our revised manuscript like below.

(the revised introduction in page 2)

<<... To overcome the first challenge, vacuum-channeled devices (vacuum field-effect transistors [VFETs]), which resemble the primitive vacuum tube triode of the early 1900s, have attracted enormous interest because they utilize ballistic transport by tunneling through the vacuum channel⁴, and recently, they demonstrate the long-term stability and the processability in 150-mm wafer scale⁵. However, industry-applicable current switching was not realized in the tunneling devices, ...>>

3. Page 3, "The ideal solution ...only one-order modulation of I_{ON}/I_{OFF} has been reported.[25] " This sentence is not proper. In reference [25], Britnell et al did not use any gate electrodes in their structure.

In the reference, Britnell et al. used silicon as a gate electrode, as shown in Fig. 1 in the reference. The gate voltage varies from 5 V to -45 V, but the drain current is modulated around 20 times at

240 K, as shown in Fig. 3B in the reference.

And we usually do not define the non-linear output curve (IV curve) as an I_{ON}/I_{OFF} ratio. The term I_{ON}/I_{OFF} is used to describe a transfer curve (or a field effect curve).

The I_{ON}/I_{OFF} ratio in our manuscript is defined by transfer curves, which is demonstrated in Fig. 1g. Figure 1g exhibits a modulation of the drain current up to 10^6 times by varying the gate voltage. To clear the point, we have revised the abstract as follows:

(the revised abstract)

<<... This device modulates the field emission barrier height across the graphene-hexagonal boron nitride interface with I_{ON}/I_{OFF} of 10^6 obtained from the transfer curves and adjustable intrinsic gain up to 4, and exhibits unprecedented current stability in temperature range of 15-300 K. ...>>

4. It is seen that the largest V_G in Fig. 1f is about 20 V; but the largest V_G in Fig. 1g-1h is cut off at 12 V. As a reader, I'm curious to see how does the IV look like at higher V_G ?

In the original manuscript, the gate dielectric (*h*BN) of the device in Fig. 1f was thicker than that of the device in Fig. 1g; the thickness of the former is 54.4 nm while the latter is 42.8 nm. Thus, a larger gate voltage (20 V) could be applied to the former device, where makes an electric field of 0.36 V/nm. To avoid this confusion, we conducted I - V measurement with an additional device (thickness of gate dielectric is 62 nm) and have revised the Fig. 1f and Fig. 1g-1h as follows.

(the revised Fig. 1f and Fig. 1g-1h in page 20)

(the Fig. 1g-j captions in page 21)

<< **g**, I_D switching performance of the semiconductor-less device. I_{D-} and I_{G-} are drain and gate current under $V_D = -18$ V. I_{D+} and I_{G+} are drain and gate current under $V_D = 29$ V. For the negative V_G , I_{ON}/I_{OFF} above 10^6 has been achieved at 300 K. **h**, Device characteristics of the n-type FEB ($V_D > 0$) at 300 K (left) and 15 K (right). **i**, Device characteristics of the p-type FEB ($V_D < 0$) at 300 K (left) and 15 K (right). For both types, very little temperature-degradation of I_D was observed. **j**, I_D switching under $V_D = -18$ V (red) and 29 V (black) also exhibited little temperature degradation from 400 K to 15 K. >>

And how do the IV curves look like when the source-drain is negatively biased (negative V_D)? Are the IV curves symmetric with respect to positive and negative V_{DS} ?

We appreciate the reviewer's comment that helps us to improve our manuscript. To respond to the reviewer's comment, we conducted additional $I-V$ measurements under both positive and negative V_D . When the source-drain is positively biased, the electron in graphene tunnels through the graphene-hBN barrier (Φ_B) by field emission. As shown in Fig. 1c-d in the revised manuscript, the electron field emission is controlled by modulating the Φ_B . In contrast, under negative V_D , hole tunnels through the barrier (Φ'_B) by field emission and is also controlled by modulating the Φ'_B (Fig. 1e-f). The revised figure 1h and 1i show I_D-V_D characteristics of our FEB under positive and negative V_D . As can be seen, they are asymmetric because the hole tunneling barrier Φ'_B extracted from FN plot (Fig. S6) is lower than the electron tunneling barrier Φ_B , by 0.28 eV. Accordingly, we revised the figures and captions of Fig. 1c-f, h and i and added S6, as shown below.

(the Fig. 1c-f in page 20)

(the Fig. 1c-f captions in page 21)

<<c – d, Band diagrams of the FEB ($V_D > 0$) under FE-dominant (c) and DT-dominant conditions (d). e – f, Band diagrams of the FEB ($V_D < 0$) under DT-dominant (e) and FE-dominant conditions (f). The gate voltage decreases from Fig. c to f.>>

(the Fig. 1h, i in page 20)

(the Fig. 1h, i captions in page 21)

<< h, Device characteristics of the n-type FEB ($V_D > 0$) at 300 K (left) and 15 K (right). i, Device characteristics of the p-type FEB ($V_D < 0$) at 300 K (left) and 15 K (right). For both types, very little temperature-degradation of I_D was observed. >>

(the Supporting Information Fig. S6 in page 12)

(the Supporting Information Fig. S6 caption in page 12)

<<Fig. S6

Fowler-Nordheim (FN) plot behavior of FEB. The IV curves of Fig. 1h, i (15 K, $V_G=0$) were replotted with axes of $\ln(|I|/V^2)$ and $1/|V|$. The barrier height for the electron (Φ_B) and hole (Φ'_B) were extracted from each slope of a line fitted to FN plot and estimated to be 2.21 eV and 1.93 eV, respectively.>>

5. Page 5, "The FE current I_D , which increases..., can be described as follows:", please cite the original paper when writing that formula.

We thank the reviewer for the comment. We have added a reference [G.-H. Lee et al., "Electron tunneling through atomically flat and ultrathin hexagonal boron nitride," *Appl. Phys. Lett.* **99**, 243114 (2011)], and the reference numbers are updated accordingly.

(the revised Results in page 5)

... The FE current I_D , which increases exponentially as the Φ_B decreases, can be described as follows³⁵:

$$I(V) = \frac{A_{\text{eff}} q^3 m V^2}{8\pi h \Phi_b d^2 m^*} \exp\left[\frac{-8\pi\sqrt{2m^*}\Phi_b^3 d}{3hqV}\right],$$

where A_{eff} is the effective tunneling area, q is the elementary charge, m is the mass of electron or hole, m^* is the tunneling effective mass, V is the applied voltage, h is Plank's constant, and d is the tunneling distance. ...

In the process of revising the supporting information, we found that the labels of y-axis in Fig. S5a was written incorrectly. So, we corrected the Fig. S5a as follows.

(the original Fig. S5a in page 11)

(the revised Fig. S5a in page 11)

Title

Semiconductor-less Vertical Transistor with I_{ON}/I_{OFF} of 10^6

Authors

Jun-Ho Lee^{1†}, Dong Hoon Shin^{2†}, Heejun Yang^{3†}, Nae Bong Jeong¹, Do-Hyun Park¹, Kenji Watanabe⁴, Takashi Taniguchi⁴, **Eunah Kim³**, Sang Wook Lee², Sung-Ho Jhang¹, Bae Ho Park¹, Young Kuk⁵ & Hyun-Jong Chung^{1*}

Affiliations

¹Department of Physics, Konkuk University, Seoul 05029, Republic of Korea.

²Department of Physics, Ewha Womans University, Seoul 03760, Republic of Korea.

³Department of Energy Science, Sungkyunkwan University, Suwon 16419, Republic of Korea.

⁴Advanced Materials Laboratory, National Institute for Materials Science, 1-1 Namiki, Tsukuba 305-0044, Japan.

⁵Daegu Gyeongbuk Institute of Science & Technology, Daegu 42988, Republic of Korea.

*Correspondence to: hjchung@konkuk.ac.kr.

†Authors contribute equally.

Abstract

Semiconductors have long been perceived as a prerequisite for solid-state transistors. Although new switching principles for nanometer-scale devices have emerged based on the deployment of two-dimensional (2D) van der Waals heterostructures, tunneling and ballistic currents through short channels are difficult to control, and semiconducting channel materials remain indispensable for practical switching. In this study, we report a semiconductor-less solid-state electronic device that exhibits an industry-applicable switching of the ballistic current. This device modulates the field emission barrier height across the graphene-hexagonal boron nitride interface **with I_{ON}/I_{OFF} of 10^6 obtained from the transfer curves** and adjustable intrinsic gain up to 4, and exhibits unprecedented current stability in temperature range of 15-400 K. The vertical device operation can be optimized with the capacitive coupling in the device geometry. The semiconductor-less switching resolves the long-standing issue of temperature-dependent device performance, thereby extending the potential of 2D van der Waals devices to applications in extreme environments.

Main text

Introduction

Semiconductors have been indispensable to solid-state electronic devices since the first solid-state electronic device (*i.e.*, the transistor in 1947) because the channel current of the transistor must be modulated by the carrier (electron and hole) density, which relies on the bandgap of the semiconductors¹. With the rapid development of the semiconductor industry, conventional three-dimensional (3D) semiconductors (Si, GaAs, and InP) are encountering new challenges in terms of increasing further spatial resolution of the device and temperature-dependent device performances in various environments. **The enhanced electric field degrades the carrier mobility in the semiconductor channel. It is because, with the enhanced electric field, the carrier starts to scatter with optical phonon of the semiconductors and lose more of its energy, resulting in the velocity saturation¹. Also,** the carrier density or device performance depends on the temperature and, as a result, deviates from Moore's law^{2,3}. To overcome the first challenge, vacuum-channeled devices (vacuum field-effect transistors [VFETs]), which resemble the primitive vacuum tube triode of the early 1900s, **have attracted enormous interest because they utilize ballistic transport by tunneling through the vacuum channel⁴, and recently, they demonstrate the long-term stability processability in 150-mm water scale⁵.** However, industry-applicable current switching was not realized in the tunneling devices, and the source (*e.g.*, silicon) and gate (*e.g.*, indium tin oxide) currents continue to rely on thermally generated carriers, which retain most of the drawbacks associated with conventional semiconductor devices.

As an alternative, two-dimensional (2D) vertical device structures have been proposed⁶⁻⁹. Despite its unprecedentedly high room-temperature mobility¹⁰, the graphene FET (GFET) still suffers from insufficient switching ($I_{ON}/I_{OFF} \sim 10$ at room temperature) because of the absence of a bandgap¹¹. Additionally, we know that artificial bandgap opening in graphene inevitably sacrifices the mobility¹². In contrast, transition metal dichalcogenide (TMD)-based FETs have shown I_{ON}/I_{OFF} values of up to 10^8 using their bandgaps, but their carrier mobilities remain at $\sim 20\%$ of that of Si⁷.

These inherent limitations can be resolved by using vertical van der Waals heterostructures and work function modulation of graphene as a novel switching principle¹³. This principle was originally demonstrated in graphene barristors (GBs)¹⁴ and has been used in various devices containing either organic^{15–18} or inorganic^{19–25} semiconductor-graphene junctions. **In addition, bipolar junction transistor (BJT)-like devices have been also investigated, where graphene was used as a base material, thus called as graphene-base transistor^{26–30}.** The switching in such devices does not rely on the thermally generated charge of semiconductors, but semiconductors are still crucial elements required to achieve efficient switching. Thus, these 2D devices have the same limitations as conventional semiconductor devices: scattering-limited carrier mobility and temperature-dependent device performance.

The ideal solution (*i.e.*, the effective switching of ballistic transport without semiconductors) has not yet been realized; indeed, only one-order modulation of I_{ON}/I_{OFF} has been reported³¹. To control ballistic transport adequately, we considered two modes by which current can tunnel through either vacuum or insulator channels: (1) direct tunneling (DT), which most graphene tunneling devices (including ref. 31) use for switching, and (2) field emission (FE), which has been rarely explored. The DT is proportional to the density of states (DOS) of two electrodes, whereas the FE is exponentially influenced by the tunneling barrier height³². When the electric field modulates the charges on graphene, both the work function and DOS at the Fermi level of the graphene are modulated. However, the two tunneling-current modes behave differently under modulation. Although the DT current produces physically limited insufficient switching (*e.g.*, an I_{ON}/I_{OFF} of ~ 50 at room temperature) via the DOS modulation of graphene³³, the FE current can be largely modulated by an exponential function of the barrier height.

Here, we report a semiconductor-less electronic device based on a van der Waals vertical heterostructure of metal-hexagonal boron nitride (*h*BN)-graphene-*h*BN-metal (Figs. 1a–b). We selected the stacked structure as the platform for an FE tunneling current because the graphene-*h*BN

junction is the cleanest 2D semimetal-insulator system³⁴. The device mainly switches the FE current by modulating the FE barrier height; therefore, we termed the device a “field emission barristor” (FEB) (Figs. 1c–f). Based on the exponential barrier height dependence of the FE current, we achieved an I_{ON}/I_{OFF} of up to 10^6 without using semiconductors (Fig. 1g). Consequently, the switching performance of our FEB exhibited ignorable degradation at 15 K (Figs. 1h–j), a temperature at which conventional semiconductor devices cannot operate. We calculated the FE barrier height variation by work function modulation in graphene using Fowler-Nordheim (FN) plot. Moreover, the work function modulation in graphene is reliably manipulated by the capacitive coupling among the gate capacitance (C_{Gate}), tunneling-channel capacitance (C_{TC}), and quantum capacitance of graphene (C_Q) in the FEB. Consequently, the above coupling effect is universal in all 2D vertical device geometries, which implies that the optimization principle can be applied to other vertical devices to improve their performance.

Results

Transport characteristics of semiconductor-less transistor. In Fig. 1g, the FEB with a gate-*h*BN thickness (t_{Gate}) of 62 nm and a tunneling-*h*BN thickness (t_{TC}) of 64 nm shows an efficient switching (by an $I_{\text{ON}}/I_{\text{OFF}}$ of up to 10^6) without semiconductors. The channel current (I_{D}) increases exponentially by an increase of the gate bias (V_{G}). As the V_{G} increases, more electrons are accumulated on the graphene, which decreases the work function of graphene by the square root of the electron density and the tunneling barrier height (Φ_{B}) for the “on” state. The FE current I_{D} , which increases exponentially as the Φ_{B} decreases, can be described as follows³⁵:

$$I(V) = \frac{A_{\text{eff}} q^3 m V^2}{8 \pi \hbar \Phi_{\text{B}} d^2 m^*} \exp \left[\frac{-8 \pi \sqrt{2 m^* \Phi_{\text{B}}^2 d}}{3 h q V} \right],$$

where A_{eff} is the effective tunneling area, q is the elementary charge, m is the mass of electron or hole, m^* is the tunneling effective mass, V is the applied voltage, h is Plank’s constant, and d is the tunneling distance. While the Schottky current depends on the temperature, the FE current barely depends on the temperature³⁶. The above formula supports the critical device operation, the switching with an $I_{\text{ON}}/I_{\text{OFF}}$ ratio of $\sim 10^6$, presented in Fig. 1g. The performance is unique among graphene-based logic devices without semiconductors. Indeed, former graphene-based tunneling or lateral devices based on DOS-dependent channel current have $I_{\text{ON}}/I_{\text{OFF}}$ ratios of ~ 10 , which is a physical limit imposed by the fact that the charge density modulation is limited to 100 at room temperature³⁷. As V_{G} increases, the tunneling mechanism of the electrons for the gate leakage current is changed from DT to FE at $V_{\text{G}} = 14$ V or a gate field of 0.23 V/nm in a similar manner with I_{D} . The I_{G} remains less than 0.5 % of I_{D} in the V_{G} range. Minimizing the leakage effect on the I_{D} , the gate field was limited to 0.23 V/nm in our study.

A critical issue affecting semiconductor-based devices—*i.e.*, temperature-limited operation—can be resolved by our semiconductor-less ballistic device. Figures 1h–j show the temperature-independent performance of the FEB: the channel current (I_{D}) exhibits little variation at temperatures

of 15-400 K under various operating conditions. This independence is attributable to the nature of the FE tunneling. Notably, the current does not degrade even at T=15 K, at which the charge carriers of most semiconductors would be frozen². The absence of degradation is a characteristic feature of our semiconductor-less ballistic device.

The channel current in Fig. 1h, i show two domains that reflect two different tunneling mechanisms (DT and FE) depending on the drain voltage (V_D). First, ineffective gating ($I_{ON}/I_{OFF} \sim 10$) appears in the DT regime at low V_D , whereas effective gating ($I_{ON}/I_{OFF} > 10^4$) is activated at high V_D . The increase in the drain voltage converts the channel current from DT to FE, allowing the modulation of the FE current shown in Fig. 1j. The transition voltages from DT to FE under positive V_D and negative V_D decrease from 27 V ($V_G=-5$ V) to 13 V ($V_G=10$ V) and from -19 V ($V_G=5$ V) to -8 V ($V_G=-10$ V), respectively; thus, a higher V_G realizes a lower Φ_B .

The Fowler-Nordheim (FN) equation can be formulated as $\ln(I_D/V_D^2) = \alpha + \beta/V_D$, where α and β have relevance to the charge density and tunneling energy barrier, respectively. Thus, the linearity between $\ln(I_D/V_D^2)$ and $1/V_D$ confirms the FN tunneling³⁸. By assuming that the graphene is a single emitter, α and β were uniquely determined, as follows. The barrier height was obtained from the modified FE equation: $\ln(I/V^2) = \alpha + \beta/V$, where α and β are $\ln \frac{A_{eff} q^3 m}{8\pi h \Phi_B d^2 m^*}$ and $-\frac{8\pi\sqrt{2m^*}d}{3hq} \Phi_B^{\frac{3}{2}}$, respectively. First, the output characteristic of a FEB was measured for a FEB with a gate dielectric of 21.5 nm and a tunneling channel of 83.8 nm, as shown in Fig. 2a. Then, a straight line of which slope is β was obtained by replotting the output characteristic of a FEB according to the modified FE equation. β includes a parameter of the FE barrier height. Therefore, β declined with increasing V_G and the FE barrier height decreased from 2.01 eV to 1.84 eV with increasing V_G from 2 V to 8 V, as exhibited in Fig. 2b.

Optimizing device performances. Device characteristics—work function modulation of graphene, intrinsic gain, I_{ON} , delay (τ), cut-off frequency (f_T) and power-delay product (PDP)—of the novel

semiconductor-less FEB's were investigated by varying t_{Gate} and t_{TC} , where τ is a time delay required to charge gate electrode with I_{ON} , f_{T} is a figure of merit of analog transistors in terms of switching speed, and PDP is that of digital ones in terms of required energy for switching^{12,39,40}. The t_{Gate} and t_{TC} affect the amplitude of the graphene work function modulation, tunneling-barrier height and thus device performances. Firstly, the capacitive coupling governs how effectively the V_{G} accumulates charges in the graphene as observed in GFET. The capacitive coupling or quantum capacitance (C_{Q}) of the graphene in the GFET has been determined to reduce the work function modulation because the C_{Q} is serially connected to the gate capacitance C_{Gate} (Fig. 3a) and, consequently, consumes a portion of V_{G} . Therefore, the accumulated charge reduces to $C_{\text{Q}}C_{\text{Gate}}/(C_{\text{Q}} + C_{\text{Gate}})$ multiplied by the V_{G} , where the larger the C_{Gate} , the higher the effect of C_{Q} , resulting in the smaller accumulated charge on the graphene^{41,42}. However, the FEB involves a more complex network of capacitors because of the additional tunneling-channel capacitor (C_{TC}), as shown in Fig. 3b. As described in the supplementary text, the potential difference of the graphene from the Dirac point (ϕ_{gr}) in the FEB is determined by the following equation⁴³:

$$C_{\text{Gate}}V_{\text{G}} + C_{\text{TC}}V_{\text{D}} = \frac{e}{\pi} \left(\frac{e}{\hbar v_{\text{F}}} \right)^2 \phi_{\text{gr}}^2 + (C_{\text{Gate}} + C_{\text{TC}})\phi_{\text{gr}}$$

The left side of the equation is the fictitious charge (Q_{fic}) on graphene accumulated by varying both the operating conditions (V_{G} and V_{D}) and the device structures (C_{Gate} and C_{TC}); the right side demonstrates how the Q_{fic} determines ϕ_{gr} with the coupling of C_{Gate} and C_{TC} . Both the C_{TC} and C_{Gate} govern the work function shift in an identical manner⁴⁴. The work function shifts with the accumulated charge, and can be obtained for the GFET and FEB by varying the $C_{\text{total}} = C_{\text{TC}} + C_{\text{Gate}}$, as shown in Fig. 3c. A smaller C_{total} produces a larger work function modulations of graphene by the same amount of charges (x-axis). Therefore, the minimum value of the C_{total} should be targeted to improve the $I_{\text{ON}}/I_{\text{OFF}}$, and the upper limit of the shift with a fixed C_{Gate} can be determined when the C_{TC} becomes 0 (*i.e.*, the case of the GFET). The above coupling analysis is generally applicable to

other vertical devices, including field-effect tunneling transistors, vFETs, thin-film barristors and GBs, that rely on the work function modulation of graphene, as shown in Fig. S1. Furthermore, the modulation can be improved by engineering the capacitance—the dielectric constant and the thickness, as described in S3^{45,46}.

Secondly, the intrinsic gain of FEB, obtained by the ratio of transconductance (g_m) to drain conductance (g_{ds})¹², is proportional to the C_{Gate} - C_{TC} ratio (t_{TC} - t_{Gate} ratio), as shown in Fig. 3d. Since they have not reported the gain of vertically stacked devices¹³, there is some doubt that the devices could not amplify (intrinsic gain <1). However, we obtained the gains of 2.5 and 3.6 using FEBs with the C_{Gate} - C_{TC} ratios of 2.0 and 3.9, respectively. Along with the other 4 FEBs, we clarified that the intrinsic gain is proportional to the C_{Gate} - C_{TC} ratio, as exhibited in Fig. 3d. It is because the fictitious charge, which determines the work function of the graphene, is linearly related to both V_G and V_D by a coupling between the C_{Gate} and C_{TC} , described in the above equations.

Lastly, the other performances such as I_{ON} , τ , f_T and PDP, were governed by t_{TC} , as exhibited in Fig. 3e. As t_{TC} increases from 19.5 nm to 54.8 nm, I_{ON} , τ and f_T are dramatically improved (~1000 times): I_{ON} increases from 0.87 mA/cm² to 1.59 A/cm²; the τ decreases from 0.7 ms to 0.52 μ s; f_T increases from 0.21 kHz to 0.59 MHz. It is because the I_{ON} exponentially depends on the barrier height at the graphene-*h*BN junction, which decreases with t_{TC} , as shown in Fig. 3f. The energy difference between the graphene's Dirac point and the conduction band of 19.5-nm thick *h*BN is obtained as 3.2 eV using single-emitter approximation; that of 301-nm thick one decreased to 2.1 eV. It is common for electron affinity of 2D materials to decrease with their thickness^{9,47}. Therefore, they can improve the device performances by increasing the t_{TC} . However, the thicker the t_{TC} , the greater is the V_D required for field-emission from the graphene to the drain electrodes. It is why the PDP increases(or worsens) as the t_{TC} increases: as the t_{TC} increases from 19.5 nm to 54.8 nm, the PDP increases from 4.3 μ J/cm² to 10.4 μ J/cm². Moreover, the lower the barrier height, the more dominant the temperature-dependent current. For example, vertical field-effect transistors (vFETs) with

graphene-WS₂ heterostructure exhibited temperature-dependent performances: the I_{ON} increased by around 1 order, and the I_{ON}/I_{OFF} decreased by approximately 2 orders¹⁹. The dependence originates from the transport mechanism of the vFET: the thermionic emission. Therefore, when optimizing the barrier height of the semiconductor-less transistor, the upper limit is determined by the device performances— I_{ON} , τ and f_T —and the lower limit is by PDP and the thermionic emission current.

Notably, *temperature independence of the performances is the unique property of the semiconductor-less vertical transistor with field-emission current*, as shown in Fig. 1j and 3g. The most FEBs exhibited temperature-independent performances. A representative device's performances are shown in Fig. 3g (black shapes), where device parameters varied by only 1.5 % or up to 11.5 %. However, some devices such as device 2 (red shapes) exhibited a little more dependence on temperature (from 17.9 to 314 %). We understand that the temperature-dependent characteristics of the semiconductor-less devices originate from Poole-Frenkel transport mediated by intrinsic defects of hBN aggregated in its defect-rich domain⁴⁸. The analysis is described in S4.

Consequently, the result indicates that the device performances of FEB can be engineered in different ways as follows: (1) the switching of FEB is governed by the capacitive coupling. (2) The intrinsic gain is proportional to the $C_{Gate}-C_{TC}$ ratio. (3) The barrier height of graphene-hBN junction decreases with the t_{TC} (4) The thicker the t_{TC} , the better is the performance of I_{ON} , τ , f_T . At the same time, PDP is degraded and temperature-dependent portion of the current increases to induce the temperature dependence of the FEB. Notably, all the characteristics of the semiconductor-less devices with optimized barrier height are temperature-independent, unless the defect-rich domain of hBN was used⁴⁸. Therefore, an optimized device geometry (e.g., t_{TC}) is indeed a key to realize the temperature-independent transistors with industry-applicable performances.

Discussion

We report the first semiconductor-less solid-state switching device with an I_{ON}/I_{OFF} of 10^6 in which a ballistic current can be effectively modulated by electric gating; thus, the device exhibits not only adjustable gain but also unprecedented temperature-independent performances, such as I_{ON} , τ , f_T and PDP. Moreover, we clarified the role of capacitive coupling among the C_{Gate} , C_{TC} and C_Q for the modulation of the graphene work function in the vertical device geometry. In our modeling, the C_{TC} is as essential as the C_{Gate} . The capacitive coupling is universal for all vertically-stacked devices based on van der Waals heterostructures, which exploit the work function modulation of the graphene as their main switching mechanisms. Our FEB achieves industry-applicable device operations with unprecedented stability over a wide range of the temperature, which resolves the long-standing issue in conventional semiconductor-based transistors and extends the potential of 2D van der Waals devices to applications in extreme environments.

Methods

Device fabrication. Monolayer graphene and two samples of *h*BN were prepared by mechanical exfoliation method. It was verified that the graphene is monolayer by using Raman spectroscopy, and the thickness of the *h*BN was measured by Atomic Force Microscope.

To make metal/*h*BN/graphene/*h*BN/metal vertical structure, the conventional wet transfer method and dry transfer method which is called PDMS stamping were conducted^{49,50}. Firstly, the *h*BN flakes were exfoliated onto PDMS surface to find several samples of few layer *h*BN. After finding two samples of few layer *h*BN on each PDMS surface using optical microscope, one was transferred onto exfoliated monolayer graphene on SiO₂ substrate by using the PDMS stamping method, and the other one was transferred onto Au/Cr gate electrode which was deposited on 300 nm SiO₂ substrate. Secondly, the sample of *h*BN/graphene was coated with 950K PMMA C4 at 4500 rpm by using spin coater. After that, the PMMA-coated *h*BN/graphene was transferred onto the *h*BN/metal structure by using the conventional wet transfer method. Thirdly, in case that total thickness of the heterostructure was thicker than 80 nm, the metal/*h*BN/graphene/*h*BN junction was coated with the PMMA to make a PMMA bridge. The PMMA was cross-linked by exposure to an electron beam with a very high dose (15,000 $\mu\text{C}/\text{cm}^2$), and the top electrodes were deposited along the cross-linked PMMA by using electron beam lithography and electron beam evaporator.

I-V measurement. Field emission current of the device was measured in vacuum probe station and physical property measurement system (PPMS) at various temperatures with Keithley 4200.

Data availability

The authors declare that the data supporting the findings of this study are available within the paper and its supplementary information files.

References

1. S. M. Sze. *Physics of semiconductor devices*. (John Wiley & Sons, New York, ed. 2, 1981).
2. Yamanouchi, C., Mizuguchi, K. & Sasaki, W. Electric conduction in phosphorus doped silicon at low temperatures. *J. Phys. Soc. Japan* **22**, 859-864 (1967)
3. Jeong, M., Doris, B., Kedzierski, J., Rim, K. & Yang, M. Silicon device scaling to the sub-10-nm regime. *Science* **306**, 2057-2060 (2004)
4. Srisophonphan, S., Jung, Y. S. & Kim, H. K. Metal-oxide-semiconductor field-effect transistor with a vacuum channel. *Nat. Nanotechnol.* **7**, 504-508 (2012)
5. Han, J. W., Seol, M. L., Moon, D. Il, Hunter, G. & Meyyappan, M. Nanoscale vacuum channel transistors fabricated on silicon carbide wafers. *Nat. Electron.* **2**, 405-411 (2019)
6. Lin, Y. M. *et al.* Wafer-scale graphene integrated circuit. *Science* **332**, 1294-1297 (2011)
7. Radisavljevic, B., Radenovic, A., Brivio, J., Giacometti, V. & Kis, A. Single-layer MoS₂ transistors. *Nat. Nanotechnol.* **6**, 147-150 (2011)
8. Cho, S. *et al.* Phase patterning for ohmic homojunction contact in MoTe₂. *Science* **349**, 625-628 (2015)
9. Kim, H. C. *et al.* Engineering Optical and Electronic Properties of WS₂ by Varying the Number of Layers. *ACS Nano* **9**, 6854-6860 (2015)
10. Bolotin, K. I. *et al.* Ultrahigh electron mobility in suspended graphene. *Solid State Commun.* **146**, 351-355 (2008)

11. Lundstrom, M. S. Graphene: The long and winding road. *Nature Mater.* **10**, 566-567 (2011)
12. Schwierz, F. Graphene transistors. *Nat. Nanotechnol.* **5**, 487-496 (2010)
13. Geim, A. K. & Grigorieva, I. V. Van der Waals heterostructures. *Nature* **499**, 419-425 (2013)
14. Yang, H. *et al.* Graphene barristor, a triode device with a gate-controlled Schottky barrier. *Science* **336**, 1140-1143 (2012)
15. Ojeda-Aristizabal, C., Bao, W. & Fuhrer, M. S. Thin-film barristor: A gate-tunable vertical graphene-pentacene device. *Phys. Rev. B - Condens. Matter Mater. Phys.* **88**, 035435 (2013)
16. Parui, S. *et al.* Gate-controlled energy barrier at a graphene/molecular semiconductor junction. *Adv. Funct. Mater.* **25**, 2972-2979 (2015)
17. Oh, G. *et al.* Graphene/pentacene barristor with ion-gel gate dielectric: flexible ambipolar transistor with high mobility and on/off ratio. *ACS Nano* **9**, 7515-7522 (2015).
18. Moon, J. S. *et al.* Lateral graphene heterostructure field-effect transistor. *IEEE Electron Device Lett.* **34**, 1190-1192 (2013)
19. Georgiou, T. *et al.* Vertical field-effect transistor based on graphene-WS₂ heterostructures for flexible and transparent electronics. *Nat. Nanotechnol.* **8**, 100-103 (2013).
20. Parui, S. *et al.* Temperature dependent transport characteristics of graphene/n-Si diodes. *J. Appl. Phys.* **116**, 244505 (2014)
21. Tian, H. *et al.* Novel field-effect schottky barrier transistors based on graphene-MoS₂ heterojunctions. *Sci. Rep.* **4**, 5951 (2014)
22. Jeong, S. J. *et al.* Thickness scaling of atomic-layer-deposited HfO₂ films and their application to wafer-scale graphene tunnelling transistors. *Sci. Rep.* **6**, 20907 (2016)

23. Huh, W. *et al.* Synaptic Barristor Based on Phase-Engineered 2D Heterostructures. *Adv. Mater.* **30**, 1801447 (2018)
24. Hwang, H. J., Heo, S., Yoo, W. B. & Lee, B. H. Graphene-ZnO:N barristor on a polyethylene naphthalate substrate. *AIP Adv.* **8**, 015022 (2018)
25. Kim, S. Y. *et al.* Threshold Voltage Modulation of a Graphene–ZnO Barristor Using a Polymer Doping Process. *Adv. Electron. Mater.* **5**, 1800805 (2019)
26. Mehr, W. *et al.* Vertical graphene base transistor. *IEEE Electron Device Lett.* **33**, 691-693 (2012)
27. Di Lecce, V. *et al.* Graphene-Base heterojunction transistor: An attractive device for terahertz operation. *IEEE Trans. Electron Devices* **60**, 4263-4268 (2013)
28. Vaziri, S. *et al.* A graphene-based hot electron transistor. *Nano Lett.* **13**, 1435 (2013)
29. Zeng, C. *et al.* Vertical graphene-base hot-electron transistor. *Nano Lett.* **13**, 2370-2375 (2013)
30. Liu, C., Ma, W., Chen, M., Ren, W. & Sun, D. A vertical silicon-graphene-germanium transistor. *Nat. Commun.* **10**, 1-7 (2019)
31. Britnell, L. *et al.* Field-effect tunneling transistor based on vertical graphene heterostructures. *Science* **335**, 947-950. (2012)
32. Simmons, J. G. Electric Tunnel Effect between Dissimilar Electrodes Separated by a Thin Insulating Film. *J. Appl. Phys.* **34**, 2581-2590 (1963)
33. Britnell, L. *et al.* Electron tunneling through ultrathin boron nitride crystalline barriers. *Nano Lett.* **12**, 1707 (2012)
34. Xue, J. *et al.* Scanning tunnelling microscopy and spectroscopy of ultra-flat graphene on

- hexagonal boron nitride. *Nat. Mater.* **10**, 282-285 (2011)
35. Lee, G. H. *et al.* Electron tunneling through atomically flat and ultrathin hexagonal boron nitride. *Appl. Phys. Lett.* **99**, 243114 (2011)
 36. Liang, S. J., Hu, W., Di Bartolomeo, A., Adam, S. & Ang, L. K. A modified Schottky model for graphene-semiconductor (3D/2D) contact: A combined theoretical and experimental study. in *Technical Digest - International Electron Devices Meeting, IEDM* **1**, 1-14 (2017).
 37. Philip Wong, H. S. & Akinwande, D. *Carbon nanotube and graphene device physics* (Cambridge University Press, Cambridge, ed. 1, 2011)
 38. R. H. Fowler. Electron emission in intense electric fields. *Proc. R. Soc. London. Ser. A, Contain. Pap. a Math. Phys. Character* **119**, 173-181 (1928)
 39. Kim, K., Choi, J. Y., Kim, T., Cho, S. H. & Chung, H. J. A role for graphene in silicon-based semiconductor devices. *Nature* **479**, 338-344 (2011)
 40. Logoteta, D., Fiori, G. & Iannaccone, G. Graphene-based lateral heterostructure transistors exhibit better intrinsic performance than graphene-based vertical transistors as post-CMOS devices. *Sci. Rep.* **4**, 6607 (2014)
 41. Xia, J., Chen, F., Li, J. & Tao, N. Measurement of the quantum capacitance of graphene. *Nat. Nanotechnol.* **4**, 505-509 (2009)
 42. Lee, J. *et al.* Is quantum capacitance in graphene a potential hurdle for device scaling? *Nano Res.* **7**, 453-461 (2014)
 43. Please see Capacitive Coupling among C_{TC} , C_G and C_Q section in the Supplementary Material.
 44. In conventional transistors. C_{Gate} and body capacitance (C_{Body}) also exist in the Si substrate. Their turn-on state was achieved when the minority charge accumulated on the channel,

resulting in inversion. In the inversion state, C_{Body} has no role in the capacitive coupling. Therefore, C_{Gate} is the most critical capacitance, and we are less concerned about capacitive coupling.

45. Robertson, J. High dielectric constant gate oxides for metal oxide Si transistors. *Reports Prog. Phys.* **69**, 327-396 (2006)
46. Jung, J. & Macdonald, A. H. Tight-binding model for graphene π -bands from maximally localized Wannier functions. *Phys. Rev. B - Condens. Matter Mater. Phys.* **87**, 195450 (2013)
47. Eda, G. *et al.* Photoluminescence from chemically exfoliated MoS₂. *Nano Lett.* **11**, 5111-5116 (2011)
48. Chandni, U., Watanabe, K., Taniguchi, T. & Eisenstein, J. P. Evidence for Defect-Mediated Tunneling in Hexagonal Boron Nitride-Based Junctions. *Nano Lett.* **15**, 7329-7333 (2015)
49. Jiao, L. *et al.* Creation of nanostructures with poly (methyl methacrylate)-mediated nanotransfer printing. *J. Am. Chem. Soc.* **130**, 12612–12613 (2008).
50. Castellanos-Gomez, A. *et al.* Deterministic transfer of two-dimensional materials by all-dry viscoelastic stamping. *2D Mater.* **1**, 011002 (2014)

Acknowledgements

H.-J.C. acknowledges support from Samsung Electronics, and National Research Foundation of Korea (NRF) grants funded by the Korea government (MSIT) (NRF-2020R1A2C1003398), and (MOE) (NRF-2018R1D1A1B07050452). B.H.P acknowledges support from the National Research Foundation of Korea (NRF) grants funded by the Korea government (MSIP) (No. 2013R1A3A2042120). S.W.L. acknowledges support from the Basic Science Research Program (NRF-2019R1A2C1080641) through the National Research Foundation of Korea (NRF) funded by the Korea government (MSIP). K.W. and T.T. acknowledge support from the Elemental Strategy Initiative conducted by the MEXT, Japan and JSPS KAKENHI Grant Numbers JP26248061, JP15K21722 and JP25106006.

Author contributions

H.-J.C. suggested the concept of semiconductor-less vertical transistor. N.B.J. prepared monolayer graphene and few layer hBN by mechanical exfoliation. D.H.P. measured Raman spectra of graphene to determinate the layer of graphene and measured thickness of hBN by using AFM. J.-H.L. stacked the samples to make a graphene/hBN vertical transistor and measured transport characteristics of the device. J.-H.L. and D.H.S. calculated modulation of tunneling barrier height and device performance. H.-J.C., D.H.S., and J.-H.L. wrote the manuscript and all authors contributed to analyze the results of the experiment and revise the manuscript.

Competing interests

The authors declare that they have no competing interests.

Materials & Correspondence

Correspondence and material requests should be addressed to H.-J.C.

Fig. 1. Fabrication of the FEB and its semiconductor-less device characteristics. **a**, Optical microscopy image of an FEB consisting of stacked metal/*h*BN/graphene/*h*BN/metal (scale bar 20 μm). (inset) Scanning electron microscopy image of polymethyl methacrylate (PMMA) bridges, which help thin metal electrodes connect through the thick stack (scale bar: 5 μm) (for more detail, see the Methods section). **b**, Schematic diagram of the FEB applying V_D and V_G . **c – d**, Band diagrams of the FEB ($V_D > 0$) under FE-dominant (**c**) and DT-dominant conditions (**d**). **e – f**, Band diagrams of the FEB ($V_D < 0$) under DT-dominant (**e**) and FE-dominant conditions (**f**). The gate voltage decreases from Fig. c to f. **g**, I_D switching performance of the semiconductor-less device. I_{D-} and I_{G-} are drain and gate current under $V_D = -18$ V. I_{D+} and I_{G+} are drain and gate current under $V_D = 29$ V. For the negative V_G , I_{ON}/I_{OFF} above 10^6 has been achieved at 300 K. **h**, Device characteristics of the n-type FEB ($V_D > 0$) at 300 K (left) and 15 K (right). **i**, Device characteristics of the p-type FEB ($V_D < 0$) at 300 K (left) and 15 K (right). For both types, very little temperature-degradation of I_D was observed. **j**, I_D switching under $V_D = -18$ V (red) and 29 V (black) also exhibited little temperature degradation from 400 K to 15 K.

Fig. 2. Single-emitter approximation of the FE from graphene. **a**, Output characteristics of FEB were measured by varying V_D from 0 to 52 V and V_G from 2 V to 8 V. As V_G increases, turn-on voltage decreases because graphene's Fermi-level increases (barrier height decreases). **b**, The characteristics were replotted with axes of $\ln(I/V^2)$ and $1/V$. From the linear fitting of the lower part (blue dashed line for $V_G = 2$ V), barrier heights were extracted to 2.01, 1.95, 1.91 and 1.84 eV when $V_G = 2, 4, 6$ and 8 V, respectively (error bars represent standard error). The height decreases by 0.17 eV, as V_G increases from 2V to 8V.

Fig. 3. Capacitive coupling, intrinsic gain and device performances. Capacitive coupling of **a**, GFET and **b**, FEB. **c**, work function shift by varying the $C_{\text{total}} = C_{\text{TC}} + C_{\text{Gate}}$. $C_{\text{total}} = 6.9 \mu\text{F}/\text{cm}^2$ when $t_{\text{Gate}} = 1 \text{ nm}$ and $t_{\text{TC}} = 1 \text{ nm}$; $C_{\text{total}} = 0.69 \mu\text{F}/\text{cm}^2$ when $t_{\text{Gate}} = 10 \text{ nm}$ and $t_{\text{TC}} = 10 \text{ nm}$; and $C_{\text{total}} = 0.069 \mu\text{F}/\text{cm}^2$ when $t_{\text{Gate}} = 100 \text{ nm}$ and $t_{\text{TC}} = 100 \text{ nm}$. The work function modulation of GFET is the upper limit of that of the FEB. **d**, intrinsic gain (g_m/g_{ds}) by varying the $t_{\text{TC}}/t_{\text{Gate}}$. The intrinsic gain is proportional to the $t_{\text{TC}}/t_{\text{Gate}}$ (the red dotted line is for guidance). **e**, device performances when t_{TC} is 19.5 nm, 30.7 nm, 32 nm, 49 nm, 50 nm and 54.8 nm, and t_{Gate} is 27.8 nm, 42.8 nm, 33 nm, 36 nm, 52 nm and 54.4 nm, respectively. I_{ON} , $1/\tau$, f_T and PDP increase with t_{TC} . They increase to ~ 1000 times as t_{TC} increases by $\sim 35 \text{ nm}$, except for PDP. **f**, field-emission barrier height by varying t_{TC} , extracted by single-emitter approximation. The barrier height between graphene's Dirac point and the conduction band decreases as t_{TC} increases. It decreases by 1.2 eV, as t_{TC} increases from 19.5 nm to 301 nm. **g**, temperature-dependent performances of FEB. I_{ON} of the most FEBs (*e.g.* device 1, black shapes) varies only 11.5 % as temperature increases from 1.78 K to 300 K; τ does less than 2.1 %; f_T does 10.6 %; PDP does 1.5 %. In contrast, some devices such as 2 (red shapes) exhibited temperature-dependent performances: I_{ON} varies 314 %; τ does 17.9 %; f_T does 177 %; PDP does 17.9 %.

Supplementary Information

S1. Capacitive Coupling among C_{TC} , C_{Gate} , and C_Q

Here, we demonstrate the limitations of the work function modulation of graphene by the coupling effect among the tunneling-channel capacitance (C_{TC}), gate capacitance (C_{Gate}) and quantum capacitance (C_Q), as shown in Fig. S1. Capacitive coupling between C_D and C_G will be addressed, and the effect of C_Q will be added. Assuming that only C_G and C_D are serially connected by a metal electrode in which charge Q accumulates, the potential of the electrode (V_M) is determined as follows:

$$(C_{TC} + C_{Gate})V_M = C_{Gate}V_G + C_{TC}V_D + Q.$$

If $V_G > 0$, $V_D > 0$ and if the middle electrode is replaced by grounded graphene, the accumulated charge consists of electrons and can be expressed as

$$Q = -en = \frac{e}{\pi} \left(\frac{e\varphi_{gr}}{\hbar v_F} \right)^2,$$

where φ_{gr} is the potential difference of graphene from the Dirac point resulting from the charge. If the work function of the gate and drain electrode is assumed to be identical to the Dirac point, for simplicity without loss of generality, V_M can be set as equal to φ_{gr} for charge Q . Therefore, φ_{gr} can be determined as follows:

$$C_{Gate}V_G + C_{TC}V_D = \frac{e}{\pi} \left(\frac{e}{\hbar v_F} \right)^2 \varphi_{gr}^2 + (C_{TC} + C_{Gate})\varphi_{gr}.$$

The left term is the fictitious charge (Q_{fic}), which is determined by the device structure (C_{Gate} and C_{TC}) and operating conditions (V_G and V_D); the right term is the quadratic function of φ_{gr} . Therefore, varying the applied voltage of V_G or V_D resulted in a shift of φ_{gr} . Considering the small modulation of drain and gate voltages with φ_{gr} (or I_D) unchanged, the above equation can be

$$C_{Gate}dV_G + C_{TC}dV_D = 0.$$

Therefore, the gain of V_D over V_G can be estimated with the ratio of C_{Gate}/C_{TC} . To increase the gain, the thickness of the tunneling channel should increase, and that of the gate dielectric should decrease.

S2. Array approximation of field emission from graphene near Dirac point.

As shown in Fig. S2a, the FE of planar graphene could be treated as an array of numerous (FE current) emitters because electron-hole puddles exist in graphene^{1,2}. The FE current was analyzed based on such an array using Seppen-Katamuki (SK, or “intercept-slope” in Japanese) plots³ by extracting the slope and y-intercept values from the drain current using the axes $\ln(I_D/V_D^2)$ and $1/V_D$. For this analysis, the FE currents of the device measured at 15 K were plotted in Fig. S2b. Then, the SK-plots were obtained, as described in Fig. S2c.

Unlike the single-emitter assumption, which appears as a single point in the SK-plot, the linear distribution in Fig. S2c implies the log-normal distributions of the radii or heights of the emitters⁴. The absolute value of the slope in the SK-plot decreases as the gate bias increases. Since the slope in the SK-plot is proportional to $\Phi_B^{3/2}$ (ref. 5), Φ_B for electron decreases by 8 % as the gate bias increases from 0 V to 12 V. Although the Fermi level of graphene varies according to the presence of impurities and defects⁶, we can assume that the Fermi level of our device is at the Dirac point when $V_D = V_G = 0$ V because of the *h*BN-graphene-*h*BN sandwich structure. Then, we obtained that the slope of the SK-plot raised to the power of 2/3 monotonically decreases relative to the charge carrier density raised to the power of 1/2 (Fig. S2d). The linear relationship between the slope in the SK-plot and the accumulated charges in the graphene validates the analysis because both are proportional to the work function of graphene

S3. Improvement of work function modulation by adopting high-k insulators

The engineering of gate insulators is another way to improve the work function modulation of the FEB. High-dielectric constant materials, such as hafnium oxide (HfO_x) and aluminum oxide (AlO_x), can be used to accumulate more charge without breaking the insulator. Such materials have been traditionally used to accumulate an identical charge at a lower voltage⁷. To explore how to improve the switching performance by engineering a gate insulator, we calculated the graphene work function shifts by varying the high-k material and its thickness for C_{Gate} , as shown in Fig. S4. The upper left part of the graph shows the work function shift with an HfO_x gate dielectric; the lower right part of the graph shows the work function shift with either $h\text{BN}$ or SiO_2 . In both cases, $h\text{BN}$ was used as the tunneling channel. We fixed V_D at half of the breakdown voltage of the tunneling channel and changed V_G from 0 V to half of the breakdown voltage of the gate insulator. Although the calculated work function shift was 0.14 eV for SiO_2 , the shift became 0.5 eV or higher for HfO_x ⁸. Thus, the work function shift of FEB can be increased using high-k gate materials.

S4. Poole-Frenkel Transport through defects of *h*BN

Poole-Frenkel (PF) Transport is a thermally assisted electron emission under high electric field, mediated by defect states in insulators. The current depends on temperature such as $\ln \frac{I_{PF}}{V_D} \propto$

$\left(U_0 - \sqrt{\frac{eV_D}{\epsilon_r \epsilon_0}} \right) \frac{1}{k_B T}$, where I_{PF} is Poole-Frenkel current, V_D is drain voltage, U_0 is energy depth from conduction band minimum to defect level of the insulator, ϵ_r is a dielectric constant of the insulator, ϵ_0 is vacuum permittivity, k_B is Boltzmann constant, and T is temperature. The I_D of FEB was measured at the temperatures of 78, 115, 175, 225 and 275 K, by varying V_G . It depends on temperature under very high electric field, as shown in Fig. S5a.

To investigate the transport mechanism, we extracted the U_0 of each V_G from the above equation. We observed that $\ln(I_D/V_D)$ linearly depended on $1/k_B T$, and U_0 was extracted to around 0.95 eV for all V_G . Interestingly, the minimum temperature, where PF transport dominates, increases as V_G increases (Fig. S5b). When V_G was lower than 4 V, I_D was governed by the PF Process at 125 K and higher temperature. As V_G increases (Fermi-level of graphene increases), the minimum temperature of the PF process also increases. It is because FE current becomes dominant when the energy of graphene's electron becomes more significant than that of the thermally excited electron in the defect. Therefore, we conclude that the temperature-dependent current of FEB is originated from PF transport, mediated by defects in *h*BN.

Fig. S1

Capacitive coupling of various novel devices: **a**, GB, **b**, FET, and **c**, GB with TMDs

Fig. S2

Analysis of FE using an array of numerous field emitters. **a**, Field emitter array of planar graphene with electron-hole puddles, where x , y and z axes are positions in the arbitrary unit and E is the energy of an electron. Each electron puddle can be an emission center because it corresponds to a local maximum of the electron energy, *i.e.* a local minimum of the work function. **b**, The output characteristic of the FEB composed of monolayer graphene, gate- h BN (thickness : 42.8 nm), and tunneling- h BN (thickness : 30.7 nm). This graph exhibits a linear relationship with the deviations on axes of $\ln(I_D/V_D^2)$ and $1/V_D$. The linear relationship confirms the FE current; the deviation originates from the array of field emitters. **c**, SK-plot of the FE of FEBs. The linear relationship suggests that the array of emitters has an identical work function, and its shift with V_G suggests that the work function is modulated. **d**, Work function modulation according to the square root of the charge. The

linear relationship confirms that the work function shift of graphene originates from the accumulated charge on the graphene, where the charge is extracted from V_G .

Fig. S3

Variation of drain and gate current with gate voltage for FEB composed of monolayer graphene, gate-*h*BN (thickness : 54.4 nm), and tunneling-*h*BN (thickness : 54.8 nm). In the FEB, Both of drain and gate current increase with gate voltage because the gate voltage affects them in different ways. In case of the drain current, it depends on modulation of tunneling barrier height caused by charge accumulation on graphene under the gate voltage and instantly increases with the gate voltage. In contrast, the gate current is mainly affected by electric field applied to the gate-*h*BN, and this makes the gate current start to increase at threshold gate voltage where the gate current is converted from DT to FE. Therefore, when the gate voltage was applied to 20 V near its threshold (17 V), the drain current was 120 times higher than the gate current as shown in Fig. S3.

Fig. S4

Work function shift with various thickness of *h*BN (lower right) or hafnium oxide (upper left) as gate dielectric, when V_D is applied half of breakdown voltage of tunneling channel and V_G spans from 0 V to half of the breakdown voltage of the gate dielectric.

Fig. S5

Poole-Frenkel Transport through defects of *h*BN. **a**, Transfer characteristic curves were measured with the temperature range of 78–275 K. The I_D of FEB under very high electric field increases by 2 orders, as the temperature increases. **b**, U_0 was extracted for each V_G using the Poole-Frenkel model, and the minimum temperature, where the transport was applicable, was also obtained. For the former, the extracted U_0 was around 0.97 eV for all V_G . For the latter, the minimum temperature increases with V_G ; with V_G less than 4 V, PF model applied to the temperature higher than 125 K; with V_G greater than 10 V, the minimum temperature increased to 225 K.

Fig. S6

Fowler-Nordheim (FN) plot behavior of FEB. The IV curves of Fig. 1h, and i (15 K, $V_G=0$) were replotted with axes of $\ln(|I|/V^2)$ and $1/|V|$. The barrier height for the electron (ϕ_B) and hole (ϕ_B') were extracted from each slope of a line fitted to FN plot and estimated to be 2.21 eV and 1.93 eV, respectively.

References

1. Xue, J. *et al.* Scanning tunnelling microscopy and spectroscopy of ultra-flat graphene on hexagonal boron nitride. *Nat. Mater.* **10**, 282-285 (2011)
2. Martin, J. *et al.* Observation of electron-hole puddles in graphene using a scanning single-electron transistor. *Nat. Phys.* **4**, 144-148
3. Ishikawa, J. *et al.* Estimation of metal-deposited field emitters for the micro vacuum tube. *Jpn. J. Appl. Phys.* **32**(3A), L342 (1993)
4. Persaud, A. Analysis of slope-intercept plots for arrays of electron field emitters. *J. Appl. Phys.* **114**, 154301 (2013)
5. Gotoh, Y., Tsuji, H. & Ishikawa, J. Relationships among the physical parameters required to give a linear relation between slope and intercept of Fowler-Nordheim plots. *Ultramicroscopy* **89**, 63-67 (2001)
6. Xia, J., Chen, F., Li, J. & Tao, N. Measurement of the quantum capacitance of graphene. *Nat. Nanotechnol.* **4**, 505 (2009)
7. Robertson, J. High dielectric constant gate oxides for metal oxide Si transistors. *Reports Prog. Phys.* **69**, 2 (2006)
8. Jung, J. & Macdonald, A. H. Tight-binding model for graphene π -bands from maximally localized Wannier functions. *Phys. Rev. B - Condens. Matter Mater. Phys.* **87**, 195450 (2013)

REVIEWER COMMENTS

Reviewer #1 (Remarks to the Author):

Let me thank the authors for their serious responses. I have no further questions.

Reviewer #2 (Remarks to the Author):

The authors addressed some of the concerns of the referees, the major messages in the manuscript are clearer than the previous version. However, there are still a few points that have to be clarified in the current form.

The authors emphasize that their novelty mainly lies in the high on/off ratio of the tunneling transistor which works in the FE regime (as compared to those reported in the DT regime). After carefully reading all the responses by the authors, I tend to believe that I may have misunderstood a bit in my previous round of reading. I would like to learn more details of the following:

1. The 2nd question raised by the other referee points to a very important thing: the leakage current of the gate start to increase a lot at some point, especially for the positive V_g . This reminds one that, as the authors claimed in their paper $V_{ds}/\text{thickness-of-BN}$ is larger than 0.35V/nm , the tunneling BN dielectric channel in between S and D may break down at that high electric field. Could the authors comment on this issue? The breakdown of gate-BN can be monitored via I_g , but how to distinguish the 'leakage current' and 'FE-current' of the channel-BN?

2. Can the authors also put a few comments: what's the difference between FE through a dielectric, and FE through vacuum?

3. In the new Fig. 1c-f, the authors use $V_d > 0$ and $V_d < 0$ as an indication for the Fermi level shift in the schematics. The authors should make a clear point why the V_d can be the sole reason to change the Fermi level, and leading to the change of carrier type? Otherwise, this schematic picture is only confusing.

4. "our FEB does not demonstrate the ambipolar behaviors, because the current is determined by the tunneling barrier height (rather than the density of states in graphene) which monotonically decreases with the charge, either electron or hole, accumulated on graphene."

This is puzzling and hardly makes any sense, at least according to the figures given in the paper in the current version. I wish I was not able to appreciate this point of the authors.

Let's put it this way: when the V_d is fixed, what is the gate doing? Isn't that the gate can still change the Fermi level of the graphene?

"The current is determined by the tunneling barrier height (rather than the density of states in graphene) which monotonically decreases with the charge, either electron or hole, accumulated on graphene."

And what determines whether electron or hole should be accumulated on the graphene?? It's the gate (well, V_d could also since the structure is quite symmetric Metal-BN-G-BN-Metal, but the V_d is fixed), right? Then there should always be a crossing point where e-h shifting takes place, thus a crossover from n-type to p-type, no?

I would agree to accept this manuscript if the authors can provide a color-mapping of IV curves

versus Gate voltage in the full parameter space (i.e., I_V from -20 to 20, V_g from -10 to 20) in both linear and log scale, along with the mapping of the gate leakage current. So that all the electron-hole questions will clear.

We appreciate the comments from the reviewers. Below are point-by-point responses to the reviewers' comments.

Reviewer #1 (Remarks to the Author):

Let me thank the authors for their serious responses. I have no further questions.

We thank the reviewer for the valuable comments.

Reviewer #2 (Remarks to the Author):

The authors addressed some of the concerns of the referees, the major messages in the manuscript are clearer than the previous version. However, there are still a few points that have to be clarified in the current form.

The authors emphasize that their novelty mainly lies in the high on/off ratio of the tunneling transistor which works in the FE regime (as compared to those reported in the DT regime). After carefully reading all the responses by the authors, I tend to believe that I may have misunderstood a bit in my previous round of reading. I would like to learn more details of the following:

1. The 2nd question raised by the other referee points to a very important thing: the leakage current of the gate start to increase a lot at some point, especially for the positive V_g . This reminds one that, as the authors claimed in their paper $V_{ds}/\text{thickness-of-BN}$ is larger than 0.35V/nm , the tunneling BN dielectric channel in between S and D may break down at that high electric field. Could the authors comment on this issue? The breakdown of gate-BN can be monitored via I_g , but how to distinguish the 'leakage current' and 'FE-current' of the channel-BN?

The authors appreciate the reviewer's comment on the distinction between the leakage current and FE-current through the channel-*h*BN in our FEB.

In our study, preventing the *h*BN from electrical breakdown was done with great care. The breakdown causes a leakage current through the channel-*h*BN, as the reviewer commented. Figure R1 below shows two cases of electrical breakdown. The current (*I*) - electric field (*E*) characteristics of two FEB devices, consisting of two different qualities of *h*BN (low- and high-quality *h*BN flakes) are demonstrated. When the breakdown occurs, the drain currents suddenly increase up to $10 \sim 10^4$ times at breakdown fields (0.62 V/nm for the low-quality *h*BN, 0.82 V/nm for the high-quality one) in the two FEB devices. However, once the breakdown occurs, the next measurement of the drain current produces a sudden increase at a low electric field. This is due to the conducting paths already formed inside the *h*BN by the breakdown. Therefore, to distinguish the leakage and FE currents through the channel *h*BN, we monitored any possible absurd increases in the drain current, and checked the reduction of the required electric field for (apparent) FE currents.

Figure R1. Electrical breakdown I-E curve of FEB devices which consist of different quality *h*BN. A breakdown strength is estimated to be 0.62 V/nm for low quality *h*BN and 0.82 V/nm for high quality *h*BN.

2. Can the authors also put a few comments: what's the difference between FE through a dielectric, and FE through vacuum?

The authors thank the reviewer for the valuable comment. As mentioned by the reviewer, our FEB could be also implemented by using vacuum tunneling barrier. Replacing the *h*BN by vacuum has pros and cons (i.e., the differences mentioned by the reviewer) as described below.

- 1) The field-emission barrier height in FEB increases to ~ 4.5 eV by the vacuum tunneling barrier, which requires higher drain voltages to overcome the tunneling barrier.
- 2) The stability could be improved by using vacuum barrier for FEB. There would be no dielectric breakdown in vacuum tunneling barrier, which could enhance the reliability of our FEB.

Thus, we put the above aspects in our manuscript in page 4.

(the revised introduction in page 4)

<< We selected the stacked structure as the platform for an FE tunneling current because the graphene-*h*BN junction is the cleanest 2D semimetal-insulator system³⁴. **While a vacuum could be another candidate for the tunneling barrier of semiconductor-less devices without a dielectric breakdown, it would require a higher operating voltage to overcome the vacuum's barrier height to switch the current.** The device mainly switches the FE current by modulating the FE barrier height; therefore, we termed the device a “field emission barristor” (FEB) (Figs. 1c–f). >>

3. In the new Fig. 1c-f, the authors use $V_d > 0$ and $V_d < 0$ as an indication for the Fermi level shift in the schematics. The authors should make a clear point why the V_d can be the sole reason to change the Fermi level, and leading to the change of carrier type? Otherwise, this schematic picture is only confusing.

We agreed with the review about the schematic figure. Thus, we have modified Figs. 1c-f and their captions.

The new schematics demonstrate that the V_D determines the type of transport (i.e., field emission or direct tunneling) with the given Fermi level of graphene (which is modulated by V_G). The key point of the FEB device is that we control the barrier height between the graphene and the channel-*h*BN by modulating the V_G .

When a positive V_D is fixed (Fig. 1c, d), the drain current I_D by electron tunneling (from the graphene to the drain metal) increases by reducing the electron-barrier height (Φ_b). However, the I_D by hole tunneling (from the drain metal to the graphene) does not increase because the Fermi level of the metal cannot be changed by V_G ; the barrier height between the drain metal and channel-*h*BN cannot be changed in our FEB.

When a negative V_D is fixed (Fig. 1e, f), the I_D by hole tunneling (from the graphene to the drain metal) increases by reducing the hole-barrier height Φ'_b . However, the I_D by electron tunneling (from the drain metal to the graphene) does not increase due to the reason mentioned above. Therefore, in the FEB, the Fermi level of graphene is controlled by V_G while the carrier type is determined by V_D (i.e., electron tunneling with a positive V_D and hole tunneling with a negative V_D). To demonstrate the above point clearer, we have revised the Figs. 1 c-f and their captions as follows:

(the Fig. 1c-f in page 20)

(the Fig. 1c-f captions in page 21)

<<c – d, Band diagrams of the FEB ($V_D > 0$) under FE-dominant (c) and DT-dominant conditions (d). The V_G modulates the accumulation of electrons on the graphene. e – f, Band diagrams of the FEB ($V_D < 0$) under DT-dominant (e) and FE-dominant conditions (f). The V_G modulates the accumulation of holes on the graphene. The gate voltage decreases from Fig. c to f.>>

4. "our FEB does not demonstrate the ambipolar behaviors, because the current is determined by the tunneling barrier height (rather than the density of states in graphene) which monotonically decreases with the charge, either electron or hole, accumulated on graphene."

This is puzzling and hardly makes any sense, at least according to the figures given in the paper in the current version. I wish I was not able to appreciate this point of the authors.

Let's put it this way: when the V_D is fixed, what is the gate doing? Isn't that the gate can still change the Fermi level of the graphene?

We note that the graphene is still ambipolar, but the tunneling in our FEB occurs in an asymmetric structure (i.e., graphene/*h*BN/metal). The asymmetric tunneling structure produces the tunneling transport only by a single type of carriers (either electrons or holes). The determination of the carriers is explained in detail in the comment '3'.

When $V_D > 0$, the I_D increases as the V_G increases (the black plots in Figs. 1g and 1j); however, when $V_D < 0$, the I_D decreases as the V_G increases (the red plots in the Figs. 1g and 1j). Unlike the graphene FET where both electrons and holes can be majority carriers according to the V_G even with a given V_D , only a single polarity of carriers is chosen as major carriers in our FEB (as explained the comment '3'). Thus, we claim that our FEB is not an ambipolar device.

"The current is determined by the tunneling barrier height (rather than the density of states in graphene) which monotonically decreases with the charge, either electron or hole, accumulated on graphene."

And what determines whether electron or hole should be accumulated on the graphene?? It's the gate (well, V_d could also since the structure is quite symmetric Metal-BN-G-BN-Metal, but the V_d is fixed), right? Then there should always be a crossing point where e-h shifting takes place, thus a crossover from n-type to p-type, no?

Yes, there should be a crossing point determined by the V_G and the initial doping-level of the graphene induced by substrates. In the *h*BN-graphene-*h*BN structure, the substrate-doping effect is ignorable [J. Xue et al., *Nature Materials* **10**, 282 (2011), H. Wang et al., *IEEE Electron Device Lett.* **32**(9), 1209-1211(2011)]. Therefore, the crossing point occurs around $V_G = 0$ V. Thus, the electron is accumulated when $V_G > 0$, and the electron barrier becomes lowered as V_G increases. The hole is accumulated when $V_G < 0$, and the hole barrier becomes lowered as V_G decreases.

Therefore, only one direction of tunneling (from the graphene to the drain metal) can be achieved for the I_D and the polarity of the carriers can be switched in the FEB device. The tunneling barrier in the direction from the drain metal to graphene cannot be modulated in our FEB. In addition, in most operating conditions, especially in the high-current regime, there cannot be a crossing point, which is distinguished from typical graphene FETs.

I would agree to accept this manuscript if the authors can provide a color-mapping of IV curves versus Gate voltage in the full parameter space (i.e., IV from -20 to 20, V_g from -10 to 20) in both linear and log scale, along with the mapping of the gate leakage current. So that all the electron-hole questions will clear.

To clarify the carrier polarity issue, we have conducted additional I - V measurements with a new device (thicknesses of channel- h BN and gate- h BN are 47 nm and 41 nm, respectively). In the following color-mappings of I_D , the V_G ranges from -10 V to 10 V and the V_D ranges from -18 V to 18 V. We confirmed that our FEB device cannot demonstrate ambipolar behaviors. We have added the linear-scaled graphs as Fig. S7.

Figure R2. Color mapping of I_D of FEB device as function of V_G and V_D on a log and linear scale.

Figure R3. Color mapping of I_G of FEB device as function of V_G and V_D on a log and linear scale.

(the Supporting Information Fig. S7 in page 13)

(the Supporting Information Fig. S7 caption in page 13)

<<Fig. S7

Color mapping of the I_D and I_G of FEB device with channel-*h*BN 47 nm and gate-*h*BN 41 nm, as a function of V_D and V_G in a linear scale. Electron current is modulated under positive V_D , and hole current is under negative V_D .>>

REVIEWERS' COMMENTS

Reviewer #2 (Remarks to the Author):

The authors have addressed my concerns. I have no further questions.